# Tight Mutual Information Estimation With Contrastive Fenchel-Legendre Optimization

**Qing Guo**[1,†]**, Junya Chen**[2]**, Dong Wang**[2]**, Yuewei Wang**[2]**, Xinwei Deng**[1]
**Lawrence Carin**[2,3]**, Fan Li**[2]**, Jing Huang**[4]**, Chenyang Tao**[2,4,†]
[1]Virginia Tech [2]Duke University [3]KAUST [4]Amazon
qguo0701@vt.edu, chenyang.tao@duke.edu

## Abstract

Successful applications of InfoNCE (Information Noise-Contrastive Estimation) and its variants have popularized the use of contrastive variational mutual information (MI) estimators in machine learning. While featuring superior stability, these estimators crucially depend on costly large-batch training, and they sacrifice bound tightness for variance reduction. To overcome these limitations, we revisit the mathematics of popular variational MI bounds from the lens of unnormalized statistical modeling and convex optimization. Our investigation yields a new unified theoretical framework encompassing popular variational MI bounds, and leads to a new simple and powerful contrastive MI estimator we name FLO. Theoretically, we show that the FLO estimator is tight, and it converges under stochastic gradient descent. Empirically, the FLO estimator overcomes the limitations of its predecessors and learns more efficiently. The utility of FLO is verified using extensive benchmarks, and we further inspire the community with novel applications in meta-learning. Our presentation underscores the foundational importance of variational MI estimation in data-efficient learning.

## 1 Introduction

Assessing the dependence between pairs of variables is integral to many scientific and engineering endeavors [66, 67]. *Mutual information* (MI) is a popular metric to quantify generic associations [50], and its empirical estimators have been widely used in applications such as independent component analysis [3], fair learning [32], neuroscience [58], Bayesian optimization [43], among others. Notably, the recent advances in deep *self-supervised learning* (SSL) heavily rely on nonparametric MI optimization [76, 57, 35, 15, 29]. In this study we investigate the likelihood-free variational approximation of MI using only paired samples, and improve the data-efficiency of current machine learning practices.

MI estimation has been extensively studied [6, 51, 50, 59, 61, 77, 10]. While most classical estimators work reasonably well for low-dimensional cases, they scale poorly to big datasets: naïve density-based estimator(s) and $k$-nearest neighbor estimators [45, 60, 24] struggle with high-dimensional inputs, while kernel estimators are slow, memory demanding and sensitive to hyperparameters [28, 27]. Moreover, these estimators are usually either non-differentiable or need to hold all data in memory. Consequently, they are not well suited for emerging applications where the data representation needs to be differentiably optimized based on small-batch estimation of MI [39]. Alternatively, one can approach MI estimation through an estimated likelihood ratio [71, 39], but the associated numerical instability has raised concerns [2].

To scale MI estimation to the growing size and complexity of modern datasets, and to accommodate the need for representation optimization [8], variational objectives have been widely utilized recently [57]. Instead of directly estimating data likelihoods, density ratios, or the corresponding gradients [81], variational approaches appeal to mathematical inequalities to construct tractable lower or

upper bounds of the mutual information [62], facilitated by the use of auxiliary critic functions[1]. This practice turns MI estimation into an optimization problem. Prominent examples include the *Barber-Agakov* (BA) estimator [4], the *Donsker-Varadhan* (DV) estimator [19], and the *Nguyen-Wainwright-Jordan* (NWJ) estimator [55]. These variational estimators are closely connected to the variational objectives for likelihood inference [1].

Despite reported successes, these variational estimators have a major limitation: their estimation variance grows exponentially to the ground-truth MI [52]. This is especially harmful to applications involving deep neural nets, as it largely destabilizes training [69]. An effective fix is to leverage multi-sample contrastive estimators, pioneered by the work of InfoNCE [57]. However, the massive reduction in the variance comes at a price: the performance of the InfoNCE estimator is upper bounded by $\log K$, where $K$ is the number of samples used for estimation (*i.e.*, batch-size) [62]. For a large MI, $K$ needs to be sufficiently large to allow for an adequate estimate, consequently placing a significant burden on computation and memory. While variants of InfoNCE have been motivated to achieve more controllable bias and variance tradeoffs [62, 69], little research has been conducted on the cost-benefit aspect of contrastive learning.

A critical insight enabled by InfoNCE is that mutual information closely connects to contrastive learning [33, 57]. Paralleled by the empirical successes of instance discrimination-based self-supervision [53, 83, 15, 35] and multi-view supervision [75, 63], InfoNCE offers an *InfoMax* explanation to why the ability to discriminate naturally paired *positive* instances from the randomly paired *negative* instances leads to universal performance gains in these applications [47, 68, 62]. Despite these encouraging developments, the big picture of MI optimization and contrastive learning is not yet complete: ($i$) There is an ongoing debate about to what extent MI optimization helps to learn useful representation [79]; ($ii$) how does the contrastive view reconcile with those non-contrastive MI estimators; and crucial for practical applications, ($iii$) are the empirical tradeoffs made by estimators such as InfoNCE absolutely necessary? Also theoretically, ($iv$) formal guarantees on the statistical convergence of popular variational non-parametric MI estimation are missing currently.

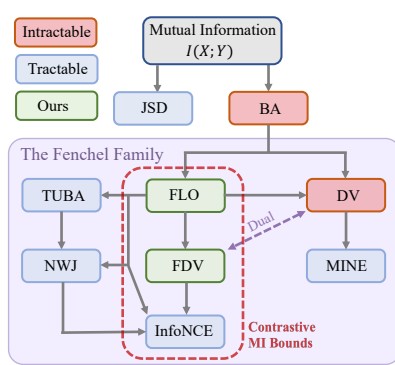

Figure 1: Schematic of variational lower bounds of mutual information. FLO provides a novel unified framework to analyze contrastive MI bounds.

In this work we seek to bridge the above gaps by approaching the MI estimation from the novel perspective of energy modeling. While this subject has recently been studied extensively using information-theoretic and variational inequalities, we embrace a new view from the lens of unnormalized statistical modeling. Our main contributions include:

- Unifying popular variational MI bounds under unnormalized statistical modeling;
- Deriving a simple but powerful novel contrastive variational bound called FLO;
- Providing theoretical justification of the FLO bound (tightness and convergence);
- Demonstrating strong empirical evidence of the superiority of FLO over its predecessors.
- Highlighting the importance of MI in data-efficient learning with novel applications

We contribute in-depth discussion to bridge the gaps between contrastive learning and MI estimation, along with principled practical guidelines informed by theoretical insights.

## 2 Fenchel-Legendre Optimization for Mutual Information Estimation

### 2.1 Preliminaries

This section briefly reviews the mathematical background needed for our subsequent developments.

**Unnormalized statistical modeling** defines a rich class of models of general interest. Specifically, we are interested in problems for which the system is characterized by an energy function $\tilde{p}_\theta(x) = \exp(-\psi_\theta(x))$, where $\theta$ is the system parameters and $\psi_\theta(x)$ is known as the *potential function*. The goal is to find a solution that is defined by a normalized version of $\tilde{p}_\theta(x)$, *i.e.*,

---

[1]When estimates are sharp, these critic functions usually recover some transformation of the likelihood ratio.

$\min_\theta \left\{ \mathcal{L}\left( \frac{\tilde{p}_\theta}{\int \tilde{p}_\theta(x')\,\mathrm{d}\mu(x')} \right) \right\}$, where $\mathcal{L}(\cdot)$ is the loss function, $\mu$ is the base measure on $\mathcal{X}$ and $Z(\theta) \triangleq \int \tilde{p}_\theta(x')\,\mathrm{d}\mu(x')$ is called the *partition function* for $\tilde{p}_\theta(x)$. Problems in the above form arise naturally in statistical physics [65], Bayesian analysis [9], and maximal likelihood estimation [72]. A major difficulty with unnormalized statistical modeling is that the partition function $Z(\theta)$ is generally intractable for complex energy functions [2], and in many applications $Z(\theta)$ is further composed by $\log Z(\theta)$, whose concavity implies any finite sample estimate Monte-Carlo of $Z(\theta)$ will render the loss function biased [64, 87]. Bypassing the difficulties caused by the intractable partition function is central to unnormalized statistical modeling [26, 54, 37, 40, 33].

**Mutual information and unnormalized statistical models.** As a generic score assessing the dependency between two random variables $(X, Y)$, *mutual information* is formally defined as the *Kullback-Leibler divergence* (KL) between the joint distribution $p(x, y)$ and product of the respective marginals $p(x)p(y)$ [67], *i.e.*, $I(X; Y) \triangleq \mathbb{E}_{p(x,y)}\left[\log \frac{p(x,y)}{p(x)p(y)}\right]$. The integrand $\log \frac{p(x,y)}{p(x)p(y)}$ is often known as the *point-wise mutual information* (PMI) in the literature. Mutual information has a few appealing properties: $(i)$ it is invariant wrt invertible transformations of $x$ and $y$, and $(ii)$ it has the intuitive interpretation of the reduced uncertainty of one variable given another variable[3].

To connect MI to unnormalized statistical modeling, we consider the classical *Barber-Agakov* (BA) estimator of MI [5]. To lower bound MI, BA introduces a variational approximation $q(y|x)$ for the posterior $p(y|x)$, and by rearranging the terms we obtain an inequality

$$
\begin{aligned}
I(X; Y) &= \mathbb{E}_{p(x,y)}\left[\log \frac{p(y|x)}{p(y)}\right] = \mathbb{E}_{p(x,y)}\left[\log \frac{q(y|x)}{p(y)}\right] + \mathbb{E}_{p(x)}[\mathrm{KL}(p(y|x) \| q(y|x))] \\
&\geq \mathbb{E}_{p(x,y)}\left[\log \frac{q(y|x)}{p(y)}\right] \triangleq I_{\texttt{BA}}(X; Y|q).
\end{aligned}
\tag{1}
$$

Here we have used notation $I_{\texttt{BA}}(X; Y|q)$ to highlight the dependence on $q(y|x)$, and when $q(y|x) = p(y|x)$ this bound is sharp. Unfortunately, this naïve BA bound is not useful for sample-based MI estimation, as we do not know the ground-truth $p(y)$. But we can bypass this difficulty by setting $q_\theta(y|x) = \frac{p(y)}{Z_\theta(x)} e^{g_\theta(x,y)}$, where we call $e^{g_\theta(x,y)}$ the *tilting function* and recognize $Z_\theta(x) = \mathbb{E}_{p(y)}[e^{g_\theta(x,y)}]$ as the associated partition function. Substituting this $q_\theta(x|y)$ into (1) gives the following *unnormalized BA* bound (UBA) that pertains to unnormalized statistical modeling [62]

$$
I_{\texttt{UBA}}(X; Y|g_\theta) \triangleq \mathbb{E}_{p(x,y)}[g_\theta(x, y) - \log Z_\theta(x)] = \mathbb{E}_{p(x)}\left[\mathbb{E}_{p(y|x)}\left[\log \frac{e^{g_\theta(x,y)}}{Z_\theta(x)}\right]\right].
\tag{2}
$$

While this UBA bound remains intractable, now with $Z_\theta(x)$ instead of $p(y)$ we can apply different techniques for empirical estimates of $Z_\theta(x)$ to render a tractable surrogate target. This has led to various popular MI bounds listed in Table 1 (see Appendix A for derivations).

**InfoNCE and noise contrastive estimation.** InfoNCE is a multi-sample mutual information estimator proposed in [57], built on the idea of *noise contrastive estimation* (NCE) [33]. NCE learns statistical properties of a target distribution by comparing the *positive* samples from the target distribution to the "*negative*" samples from a carefully crafted noise distribution, and this technique is also known as *negative sampling* in some contexts [53, 30]. The InfoNCE estimator implements this contrastive estimation idea via using the naïve empirical estimate of $Z_\theta(x)$ in UBA[4], *i.e.*

$$
I_{\texttt{InfoNCE}}^K(X; Y|g_\theta) \triangleq \mathbb{E}_{p^K(x,y)}\left[\log \frac{e^{g_\theta(x_1,y_1)}}{\frac{1}{K}\sum_j e^{g_\theta(x_1,y_j)}}\right], I_{\texttt{InfoNCE}}^K(X; Y) \triangleq \max_{g_\theta \in \mathcal{F}}\{I_{\texttt{InfoNCE}}^K(X; Y|g_\theta)\},
\tag{3}
$$

where $g_\theta$ is known as the *critic* in the nomenclature of contrastive learning, and we have used $p^K(x, y)$ to denote $K$ independent draws from the joint density $p(x, y)$, and $\{(x_k, y_k)\}_{k=1}^K$ for each pair of samples. Here the positive and negative samples are respectively drawn from the joint $p(x, y)$ and product of marginals $p(x)p(y)$. Intuitively, InfoNCE tries to accurately classify the positive samples when they are mixed with negative samples, and the Proposition below formally characterizes InfoNCE's statistical properties as a MI estimator.

---

[2]In the sense that they do not render closed-from expressions.

[3]Formally, $I(X; Y) = H(X) - H(X|Y) = H(Y) - H(Y|X)$, where $H(X)$ (resp. $H(X|Y)$) denotes the Shannon entropy (resp. conditional Shannon entropy) of a random variable.

[4]This estimator is technically equivalent to the original definition due to the symmetry of $K$ samples.

**Proposition 2.1** ([62]). `InfoNCE` is an asymptotically tight mutual information lower bound, *i.e.* $I_{\texttt{InfoNCE}}^K(X;Y|g_\theta) \leq I(X;Y)$, $\lim_{K\to\infty} I_{\texttt{InfoNCE}}^K(X;Y) \to I(X;Y)$.

**Fenchel-Legendre duality.** Our key idea is to exploit the convex duality for MI estimation. Let $f(t)$ be a proper convex, lower-semicontinuous function; then its convex conjugate function is defined as $f^*(v) \triangleq \sup_{t\in\mathcal{D}(f)}\{tv - f(t)\}$, where $\mathcal{D}(f)$ is the domain of function $f$ [38]. We call $f^*(v)$ the *Fenchel conjugate* of $f(t)$, which is also known as the *Legendre transform* in physics. The Fenchel conjugate pair $(f, f^*)$ are dual to each other, in the sense that $f^{**} = f$, *i.e.*, $f(t) = \sup_{v\in\mathcal{D}(f^*)}\{vt - f^*(v)\}$. For $f(t) = -\log(t)$ and its Fenchel conjugate $f^*(v) = -1 - \log(-v)$, we have inequality

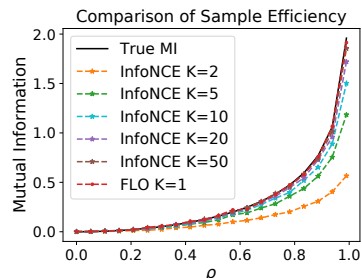

Figure 2: $K$-sample `InfoNCE` and single-sample `FLO`. Note `FLO` is tight regardless of sample-size.

$$-\log(t) \geq -u - e^{-u}t + 1, \quad \text{for } u \in \mathbb{R} \tag{4}$$

with the equality holds when $u = \log(t)$.

## 2.2 Fenchel-Legendre Optimization for tight mutual information estimation

With the above mathematical tools, we are ready to present the main result of this paper: a tight, data-efficient variational MI lower bound that can be efficiently implemented.

**Lower bounding MI with Fenchel-Legendre Optimization.** Our key insight is that MI estimation is essentially an unnormalized statistical model, which can be efficiently handled by the Fenchel-Legendre transform technique. Take the integrand from `UBA` in (2) and we can rewrite it as

$$\log \frac{\exp(g_\theta(x,y))}{Z_\theta(x)} = -\log\left\{\mathbb{E}_{p(y')}[\exp(g(x,y') - g(x,y))]\right\}, \tag{5}$$

where $p(y')$ is the same probability density as $p(y)$ (*i.e.*, $Y'$ is an independent copy of $Y$). Now let us use the Fenchel inequality of $-\log(t)$ from (4), plugging it into the above equation and then we have

$$\log \frac{\exp(g_\theta(x,y))}{Z_\theta(x)} \geq \left\{-u - e^{-u}\mathbb{E}_{p(y')}[\exp(g(x,y') - g(x,y))]\right\} + 1. \tag{6}$$

for all $u \in \mathbb{R}$. This implies for any function $u_\phi(x,y) : \mathcal{X} \times \mathcal{Y} \to \mathbb{R}$, the following inequality holds

$$\log \frac{\exp(g_\theta(x,y))}{Z_\theta(x)} \geq -\{u_\phi(x,y) + e^{-u_\phi(x,y)}\mathbb{E}_{p(y')}[\exp(g(x,y') - g(x,y))]\} + 1. \tag{7}$$

By putting (7) back to (2), we obtain our new Fenchel-Legendre Optimization (`FLO`) MI lower bound

$$I_{\texttt{FLO}}(X;Y|g_\theta, u_\phi) \triangleq \mathbb{E}_{p(x,y)}\left[-\{u_\phi(x,y) + e^{-u_\phi(x,y)}\mathbb{E}_{p(y')}[e^{g_\theta(x,y') - g_\theta(x,y)}]\}\right] + 1, \tag{8}$$

and concludes the proof for the following Proposition.

**Proposition 2.2.** $I_{\texttt{FLO}}(X;Y|g_\theta, u_\phi) \leq I_{\texttt{UBA}}(X;Y|g_\theta) \leq I(X;Y)$.

In practice, `FLO` can be estimated with the following naïve empirical $K$-sample estimator

$$\hat{I}_{\texttt{FLO}}^K(X;Y|g_\theta, u_\phi) \triangleq -\left\{u_\phi(x_i,y_i) + e^{-u_\phi(x_i,y_i)}\frac{1}{K-1}\sum_{j\neq i}e^{g_\theta(x_i,y_j) - g_\theta(x_i,y_i)}\right\} + 1. \tag{9}$$

Since the summation in $\hat{I}_{\texttt{FLO}}^K$ is not encapsulated by a convex log transformation, $I_{\texttt{FLO}}^K \triangleq \mathbb{E}_{p^K}[\hat{I}_{\texttt{FLO}}^K]$ is an unbiased estimator for $I_{\texttt{FLO}}(X;Y|g_\theta, u_\phi)$ independent of the batch size $K$ (see Figure 2).

**Why is the `FLO` bound more appealing?** At first sight, it may appear counter-intuitive that $I_{\texttt{FLO}}$ is a better MI bound compared to prior arts such as `NWJ` or `InfoNCE`: it seems to be more complicated as an extra variational function $u_\phi(x,y)$ has been introduced. To answer this question, we next explain the statistical meaning of the newly introduced $u_\phi(x,y)$, and establish some important statistical properties of `FLO` that makes it more favorable: that $I_{\texttt{FLO}}$ is tight, meaning the ground-truth MI can be recovered for some specific choice of $g_\theta(x,y)$ and $u_\phi(x,y)$; and that $I_{\texttt{FLO}}^K$ for any batch size $K$ is effectively optimizing `InfoNCE` with an infinite batch size. And in Sec 2.4, we further justify `FLO`'s advantages from optimization perspectives.

Given the close connection between FLO and UBA, we first recall UBA's optimal critic that gives the tight MI estimate is $g^*(x, y) = \log p(x|y) + c(x)$, where this $c(x)$ can be any function of $x$ [49]. This $g^*(x, y)$ is not directly meaningful in a statistical sense, however, by integrating out $y'$, we have

$$\mathbb{E}_{p(y')}\left[e^{g^*(x,y')-g^*(x,y)}\right] = \mathbb{E}_{p(y')}\left[\frac{p(x|y')}{p(x|y)}\right] = \frac{p(x)}{p(x|y)} = \frac{p(x)p(y)}{p(x,y)}, \tag{10}$$

which is the likelihood ratio between the marginals and joint. On the other hand, based on the Fenchel-Legendre inequality (4), we know for fixed $g(x, y)$ our FLO bound in (8) can be maximized with $u_g(x, y) = \log \mathbb{E}_{p(y')}\left[e^{g(x,y')-g(x,y)}\right]$. Putting these all together we have $u_{g^*}(x, y) = -\log \frac{p(x,y)}{p(x)p(y)}$. This shows the $u_\phi(x, y)$ introduced in FLO actually tries to recover the negative PMI. Comparing to the competing MI bounds that only optimizes for $g_\theta$, eliminating the drift term $c(x)$ reveals FLO enjoys the appealing *self-normalizing* property [33] that helps stabilize training. Plugging $(g^*, u_{g^*})$ into (8), we readily see $I_{\texttt{FLO}}(X; Y|u_{g^*}, g^*) = I(X; Y)$, proving FLO is a tight MI bound.

**Proposition 2.3.** The FLO estimator is tight, the eqality holds when $g(x, y) = \log p(x|y) + c(x)$ for arbitrary function $c(x)$ and $u(x, y) = -\log \frac{p(x,y)}{p(x)p(y)}$.

**Corollary 2.4.** Let $(g^*, u_{g^*})$ be the maximizers for (8), then $I(X; Y) = \mathbb{E}_{p(x,y)}[-u_{g^*}(x, y)]$.

Finally, we give a simple asymptotic argument showing FLO essentially optimizes InfoNCE with an infinite batch size. In virtue of the law of large numbers, we have the denominator in InfoNCE converging to $\lim_{K\to\infty} \frac{1}{K}\sum_{j=1}^{K} e^{g_\theta(x_i,y_j)} \to \mathbb{E}_{p(y')}[e^{g_\theta(x_i,y')}] = Z_\theta(x_i)$, and consequently it recovers the UBA bound. Since FLO is derived from UBA, we can view FLO as using the optimization of $u_\phi(x, y)$ to amortize the difficulty of evaluating infinite number of $e^{g_\theta(x_i,y_j)}$ with InfoNCE.

**Efficient implementations of** FLO. A lingering concern is that the newly introduced $u_\phi(x, y)$ can incur extra computation overhead. This is not true, as we can maximally encourage parameter sharing by jointly model $u_\phi(x, y)$ and $g_\theta(x, y)$ with a single neural network $f_\Psi(x, y) : \mathcal{X} \times \mathcal{Y} \to \mathbb{R}^2$ with two output heads, *i.e.*, $[u_i, g_i] = f_\Psi(x_i, y_i)$. Consequently, while FLO adopts a dual critics design, it does not actually invoke extra modeling cost compared to its single-critic counterparts (*e.g.*, InfoNCE). Experiments show this shared parameterization in fact promotes synergies and speeds up learning (see our ablation studies in Appendix F.6).

To further enhance the computation efficiency, we consider a massively parallelized *bi-linear* critic design that uses all in-batch samples as negatives. Let $g_\theta(x, y) = \tau \cdot \langle h_\theta(x), \tilde{h}(y) \rangle$, where $h : \mathcal{X} \to \mathbb{S}^p$ and $\tilde{h} : \mathcal{Y} \to \mathbb{S}^p$ are respectively encoders that map data to unit sphere $\mathbb{S}^p$ embedded in $\mathbb{R}^{p+1}$, $\langle a, b \rangle = a^T b$ is the inner product operation, and $\tau > 0$ is the inverse temperature parameter. Thus the evaluation of the *Gram* matrix $G = \tau \cdot h(\mathbb{X})^T \tilde{h}(\mathbb{Y})$, where $[\mathbb{X}, \mathbb{Y}] \in \mathbb{R}^{K \times (d_x + d_y)}$ is a mini-batch of $K$-paired samples and $g_\theta(x_i, y_j) = G_{ij}$, can be parallelized via matrix multiplication. In this setup, the diagonal terms of $G$ are the positive scores while the off-diagonal terms negative scores. A similar strategy has been widely employed in the contrastive representation learning literature (*e.g.*, [15])[5]. We can simply model the PMI critic as $u(x, y) = \texttt{MLP}(h(x), \tilde{h}(y))$, whose computation cost is almost neglectable in practice, where feature encoders $h, \tilde{h}$ dominate computing.

## 2.3 Connections to the existing MI bounds

Due to space limitations, we elaborate the connections to the existing MI bounds here, and have relegated an extended related work discussion in a broader context to Appendix A, H.

**From** $\log$**-partition approximation to MI bounds.** To embrace a more holistic understanding, we list popular variational MI bounds together with our FLO in Table 1, and visualize their connections in Figure 1. With the exception of JSD, these bounds can be viewed from the perspective of unnormalized statistical modeling, as they differ in how the log partition function $\log Z(x)$ is estimated. We broadly categorize these estimators into two families: the log-family (DV, MINE, InfoNCE) and the exponential-family (NWJ, TUBA, FLO). In the log-family, DV and InfoNCE are multi-sample estimators that leverage

---

[5]As an important note to the community, most open source implementations for the bilinear contrastive loss have mechanically implemented $\frac{1}{T}\langle \cdot, \cdot \rangle$ following the practice from pioneering contrastive learning studies, which is numerically unstable compared to our parameterization $\tau\langle \cdot, \cdot \rangle$ proposed here.

Table 1: Comparison of popular variational MI estimators. Here $g(x,y), u(x,y)$ and $u(x)$ are variational functions to be optimized, $\sigma(u) = \frac{1}{1+\exp(-u)}$ is the Sigmoid function, $\mathcal{E}[f(u),\eta]$ denotes exponential average of function $f(u)$ with decay parameter $\eta \in (0,1)$, and $\alpha \in [0,1]$ is the balancing parameter used by $\alpha$-`InfoNCE` trading off bias and variance between `InfoNCE` and `TUBA`. we use $(x_i, y_i)$ to denote positive samples from the joint density $p(x,y)$, and $(x_i, y_j)$ or $(x'_k, y'_k)$ to denote negative samples drawn from the product of marginal $p(x)p(y)$. In context, $y_\oplus$ and $y_\ominus$ have the intuitive interpretation of positive and negative samples. We exclude variational upper bounds here because their computations typically involve the explicit knowledge of conditional likelihoods.

| Name | Objective | Bias | Var. | Converge |
|---|---|---|---|---|
| $(x_i, y_i) \overset{iid}{\sim} p(x,y),\ (x'_k, y'_k) \overset{iid}{\sim} p(x)p(y),\ m_{\alpha,u}(x, y_{1:K}) \triangleq \alpha \frac{1}{K} \left\{ \sum_{k=1}^{K} \exp(g(x, y_k)) \right\} + (1-\alpha)\exp(u(x))$ | | | | |
| DV [19] | $g(x_i, y_i) - \log(\sum_{k=1}^{K} \exp(g(x'_k, y'_k))/K)$ | high | high | no |
| MINE [7] | $g(x_i, y_i) - \log(\mathcal{E}[\exp(g(x_i, y_j)), \eta])$ | low | high | no |
| NWJ [55] | $g(x_i, y_i) - \exp(g(x_i, y_j) - 1)$ | low | high | no |
| JSD [39] | $g^*(x_i, y_i) - \exp(g^*(x_i, y_j) - 1)$ | low | high | no |
| | $g^* \xleftarrow{\arg\max} \{\log \sigma(g(x_i, y_i)) + \log \sigma(-g(x_i, y_j))\}$ | | | |
| TUBA [62] | $g(x_i, y_i) + u(x_i) + 1 - \exp(g(x_i, y_j) - u(x_i))$ | low | high | no |
| InfoNCE [57] | $g(x_i, y_i) - \log(\sum_j \exp(g(x_i, y_j))/K)$ | high | low | no |
| $\alpha$-InfoNCE [62] | $g(x_i, y_i) - g(x_i, y_j) - \log(m_{\alpha,u}(x, y_{1:K})) + \log(m_{\alpha,u}(x'_k, y'_k))$ | | | no |
| $\alpha$-InfoNCE interpolates between low-bias high-var ($\alpha \to 1$, NWJ) to high-bias low-var ($\alpha \to 0$, InfoNCE) | | | | |
| FLO (ours) | $-u(x_i, y_i) - \exp(-u(x_i, y_i) + g(x_i, y_j) - g(x_i, y_i))$ | **low** | **moderate** | **yes** |

direct Monte-Carlo estimates $\hat{Z}$ for $\log Z(x)$, and these two differ in whether to include the positive sample in the denominator or not. To avoid the excessive in-batch computation of the normalizer and the associated memory drain, `MINE` further employed an *exponential moving average* (EMA) to aggregate the normalizer across batches. Note for the log-family estimators, their variational gaps are partly caused by the log-transformation on finite-sample average due to Jensen's inequality (*i.e.*, $\log Z = \log \mathbb{E}[\hat{Z}] \geq \mathbb{E}[\log \hat{Z}]$). In contrast, the objective of exponential-family estimators do not involve such log-transformation, since they can all be derived from the Fenchel-Legendre inequality: `NWJ` directly applies the Fenchel dual of $f$-divergence for MI [56], while `TUBA` exploits this inequality to compute the log partition $\log Z(x) = \log \mathbb{E}_{p(y')}[\exp(g(x,y'))]$. Motivated from a contrastive view, our `FLO` applies the Fenchel-Legendre inequality to the log-partition of contrast scores.

**A contrastive view for MI estimation.** The MI estimators can also be categorized based on how they contrast the samples. For instance, `NWJ` and `TUBA` are generally considered to be non-contrastive estimators, as their objectives do not compare positive samples against negative samples on the same scale (*i.e.*, $\log$ versus $\exp$), and this might explain their lack of effectiveness in representation learning applications. For `JSD`, it depends on a two-stage estimation procedure similar to that in adversarial training to assess the MI, by explicitly contrasting positive and negative samples to estimate the likelihood ratio. This strategy has been reported to be unstable in many empirical settings. The log-family estimators can be considered as a multi-sample, single-stage generalization of `JSD`. However, the `DV` objective can go unbounded thus resulting in a large variance, and the contrastive signal is decoupled by the EMA operation in `MINE`. Designed from contrastive perspectives, `InfoNCE` trades bound tightness for a lower estimation variance, which is found to be crucial in representation learning applications. Our `FLO` formalizes the contrastive view for exponential-family MI estimation, and bridges existing bounds: the PMI normalizer $\exp(-u(x,y))$ is a more principled treatment than the EMA in `MINE`, and compared to `DV` the positive and negative samples are explicitly contrasted and adaptively normalized.

**Important FLO variants.** We now demonstrate that `FLO` is a flexible framework that not only recovers existing bounds, but also derives novel bounds such as

$$I_{\text{FDV}} \triangleq \texttt{StopGrad}[I_{\text{DV}}(\{(x_i, y_i)\})] + \frac{\sum_j \exp(c_\theta(x_i, y_i, y_j))}{\texttt{StopGrad}[\sum_j \exp(c_\theta(x_i, y_i, y_j))]} - 1. \tag{11}$$

Recall the optimal $g^*(x,y) = \log p(x|y) + c(x)$ and $u^*(x,y) = -\log \frac{p(x,y)}{p(x)p(y)}$, which motivates us to parameterize $u(x,y)$ in the form of $-g_\theta(x,y) + s_\psi(x)$, where $s_\psi(x)$ models the arbitrary drift $c(x)$, and this recovers the TUBA bound. Additionally, we note that (*i*) fixing either of $u$

and $g$, and optimizing the other also gives a valid lower bound to MI; and $(ii)$ a carefully chosen multi-input $u(\{(x_i, y_i)\})$ can be computationally appealing. As a concrete example, if we set $u_\phi$ to $\mathfrak{u}_\theta(\{(x_i, y_i)\}) \leftarrow \log\left(\frac{1}{K}\sum_j e^{c(x_i, y_i, y_j; g_\theta)}\right)$ and update $u_\theta(x, y)$ while artificially keeping the critic $g_\theta(x, y)$ fixed [6], then FLO falls back to DV. Alternatively, we can consider the Fenchel dual version of it: using the same multi-input $\mathfrak{u}_\theta(\{(x_i, y_i)\})$ above, treat $u_\phi$ as fixed and only update $g_\theta$, and this gives us the novel MI objective in (11), we call it *Fenchel-Donsker-Varadhan* (FDV) estimator.

## 2.4 Gradient and convergence analysis of FLO

In this section, we will establish that FLO better optimizes the MI because its gradient is more accurate than competing variational bounds such as NWJ and TUBA; also, we provide the first convergence analysis for variational MI estimation by showing FLO converges under SGD.

First, recall most tractable variational MI bounds are derived from and upper bounded by the intractable UBA bound [62]. For instance, with the same critic $g_\theta$ we have $I_{\text{NWJ}} \leq I_{\text{TUBA}} \leq I_{\text{UBA}}$. So if we can show $\nabla_\theta I_{\text{FLO}} \approx \nabla_\theta I_{\text{UBA}}$ then FLO is better optimized. To simplify notations, we denote $c_\theta(x, y, y') \triangleq g_\theta(x, y') - g_\theta(x, y)$ and $\mathcal{E}_\theta(x, y) \triangleq 1/\mathbb{E}_{p(y')}[e^{c_\theta(x, y, y')}]$, and we can easily verify

$$\mathbb{E}_{p(y')}\left[\nabla_\theta\left\{e^{c_\theta(x, y, y')}\right\}\right] = \nabla_\theta\left\{\frac{1}{\mathcal{E}_\theta(x, y)}\right\} = -\frac{\nabla\mathcal{E}_\theta(x, y)}{(\mathcal{E}_\theta(x, y))^2} = -\frac{\nabla_\theta \log\mathcal{E}_\theta(x, y)}{\mathcal{E}_\theta(x, y)}. \quad (12)$$

Since for fixed $g_\theta(x, y)$ the corresponding optimal $u_\theta^*(x, y)$ maximizing $I_{\text{FLO}}(u_\phi, g_\theta) \triangleq 1 - \left\{u_\phi(x, y) + \mathbb{E}_{p(y')}[e^{-u_\phi(x, y) + c(x, y, y'; g_\theta)}]\right\}$ is given by $u_\theta^*(x, y) = \log\mathbb{E}_{p(y')}[e^{c_\theta(x, y, y')}] = -\log\mathcal{E}_\theta(x, y)$ (using (4)), we see that the term $e^{-u_\phi(x, y)}$ is essentially optimized to approximate $\mathcal{E}_\theta(x, y)$. To emphasize this point, we now write $\hat{\mathcal{E}}_\theta(x, y) \triangleq e^{-u_\phi(x, y)}$. When this approximation is sufficiently accurate (*i.e.*, $\mathcal{E}_\theta \approx \hat{\mathcal{E}}_\theta$), we can see that $\nabla I_{\text{FLO}}$ approximates $\nabla I_{\text{UBA}}$ as follows

$$\nabla_\theta\{I_{\text{FLO}}(u_\phi, g_\theta)\} = -\mathbb{E}_{xy}\left[e^{-u_\phi(x, y)}\mathbb{E}_{y'}[\nabla_\theta e^{c_\theta(x, y, y')}]\right] = \mathbb{E}_{xy}\left[\frac{\hat{\mathcal{E}}_\theta(x, y)}{\mathcal{E}_\theta(x, y)}\nabla_\theta\log\mathcal{E}_\theta(x, y)\right]$$

$$\approx \mathbb{E}_{xy}\left[\nabla_\theta\log\mathcal{E}_\theta(x, y)\right] = \nabla_\theta\left\{\mathbb{E}_{p(x, y)}[\log\mathcal{E}_\theta(x, y)]\right\} = \nabla_\theta\{I_{\text{UBA}}(g_\theta)\}. \quad (13)$$

We can prove FLO will converge under much weaker conditions, even when this approximation $\hat{u}(x, y)$ is rough. The intuition is as follows: in (13), the term $\frac{\hat{\mathcal{E}}_{\theta_t}}{\mathcal{E}_{\theta_t}}$ only rescales the gradient, so the optimizer is still proceeding in the same direction as UBA in SGD. The informal version of our result is summarized in the Proposition below (see Appendix E for the formal version and proof).

**Proposition 2.5** (Convergence of FLO, informal version). Let $\{\eta_t\}_{t=1}^\infty$ be the stochastic *Robbins-Monro* sequence of learning rates: $\sum_t \mathbb{E}[\tilde{\eta}_t] = \infty$ and $\sum_t \mathbb{E}[\tilde{\eta}_t^2] < \infty$. If $\frac{\hat{\mathcal{E}}_{\theta_t}}{\mathcal{E}_{\theta_t}}$ is bounded between $[a, b]$ ($0 < a < b < \infty$), then under the stochastic gradient descent scheme described in Algorithm 1, $\theta_t$ converges to a stationary point of $I_{\text{UBA}}(g_\theta)$ with probability 1, *i.e.*, $\lim_{t\to\infty} \|\nabla I_{\text{UBA}}(g_{\theta_t})\| = 0$. Additionally assume $I_{\text{UBA}}$ is convex with respect to $\theta$, then FLO converges with probability 1 to the global optimum $\theta^*$ of $I_{\text{UBA}}$ from any initial point $\theta_0$.

---

**Algorithm 1** FLO

---

Empirical data $\hat{p}_d = \{(x_i, y_i)\}_{i=1}^n$
Model parameters $\Psi = (\theta, \phi)$
**for** $t = 1, 2, \cdots$ **do**
    Sample $i, j \overset{iid}{\sim} [n]$
    $u_{ii} = u_\phi(x_i, y_i), g_{ii} = g_\theta(x_i, y_i),$
    $g_{ij} = g_\theta(x_i, y_j)$
    $\mathcal{F} = u_{ii} + \exp(-u_{ii} + g_{ij} - g_{ii})$
    $\Psi_t = \Psi_t - \eta_t \nabla_\Psi \mathcal{F}$
**end for**

---

Importantly, this marks the first SGD convergence analyses for variational MI estimators. The convergence analyses for MI estimation is non-trivial and scarce even for those standard statistical estimators [59, 24, 64]. For variational MI bounds, existing convergence analyses only apply to oracle estimators but not practical estimators discussed here [55, 70]. The lack of convergence guarantees has led to a proliferation of unstable MI-estimators used in practice (in particular, DV, JSD, and MINE) that critically rely on various empirical hacks to work well (see discussions in [69]). Our work establishes a family of variational MI estimators that provably converges, a contribution we consider significant as it fills an important gap in current literature on both theoretical and practical notes. While our convergence result is local, in practice we have observed that FLO.

---

[6]That is to say $g_\theta$ in $u_\phi$ is an independent copy of $g_\theta$.

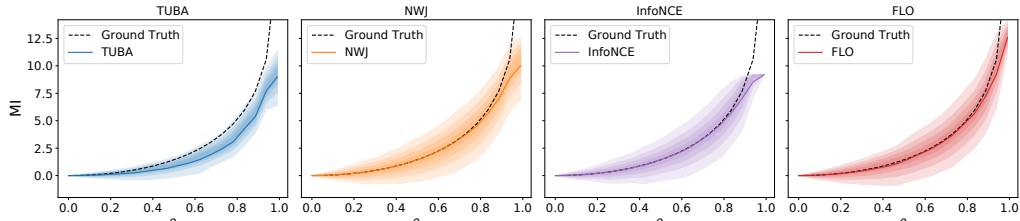

Figure 3: Bias-variance plot for popular variational MI bounds with the 10-D Gaussians. Estimators that are more concentrated around the dashed line is considered better (low-bias, low-variance). In the more challenging high-MI regime, FLO shows a clear advantage over competing alternatives, where FLO pays less price in variance to achieve even better accuracy when tight estimation is impossible.

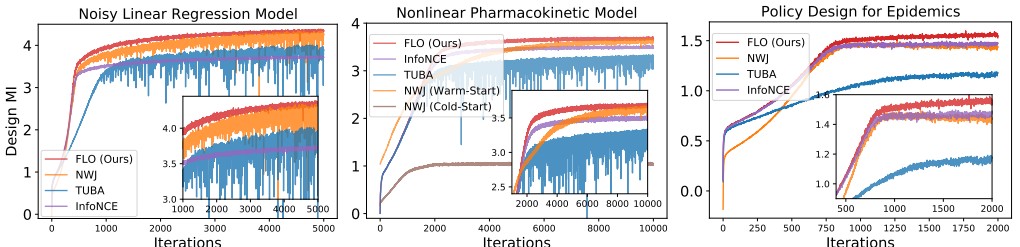

Figure 4: Bayesian Optimal Experiment Design results. FLO consistently performs best, demonstrating superior strength in learning efficiency and robustness. NWJ takes the runner-up, but it has larger variance and is sensitive to network initializations. InfoNCE is less competitive due to low sample inefficiency, but its smaller variance helps in the more challenging dynamic case.

# 3 Experiments

We consider an extensive range of tasks to validate FLO and benchmark it against state-of-the-art solutions. To underscore the practical significance of MI in efficient machine learning, we demonstrate example applications from data collection (in statistical parlance, experimental design), self-supervised pre-training, to meta/transfer-learning. Limited by space, we present only the key results in the main text, and defer ablation studies and details of our experimental setups to the Appendix. Our code is available from https://github.com/qingguo666/FLO. All experiments are implemented with PyTorch.

**Comparison to baseline MI bounds.** We start by comparing FLO to the following popular competing variational estimators: NWJ, TUBA, and InfoNCE. We use the bilinear critic implementation for all models which maximally encourages both sample efficiency and code simplicity, and this strategy does perform best based on our observations. We consider the synthetic benchmark from [62], where $(X \in \mathbb{R}^d, Y \in \mathbb{R}^d)$ is jointly standard Gaussian with diagonal cross-correlation parameterized by $\rho \in [0, 1)$. We report $d = 10$ and $\rho \in [0, 0.99]$ here (other studies only report $\rho$ up to 0.9, which is less challenging.), providing a reasonable coverage of the range of MI one may encounter in empirical settings.

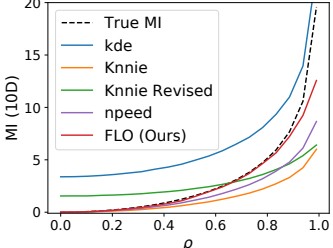

Figure 5: FLO compares favorably to classical MI estimators.

To focus on the bias-variance trade-off, we plot the decimal quantiles in addition to the estimated MI in Figure 3, where FLO significantly outperformed its variational counterparts in the more challenging high-MI regime. In Figure 5, we show FLO also beats classical MI estimators [45, 80, 25]. In the Appendix H, we further discuss recent works on parametric estimators [17, 10] and alternative information metrics [85].

**Bayesian optimal experiment design (BOED).** We next direct our attention to BOED, a topic of significant interest shared by the statistical and machine learning communities [11, 82, 36, 23]. The performance of machine learning models crucially relies on the quality of data supplied for training, and BOED is a principled framework that optimizes the data collection procedure (in statistical parlance, conducting *experiments*) [22]. Mathematically, let $x$ be the data to be collected, $\theta$ be the parameters to be inferred, and $d$ be the experiment parameters the investigator can manipulate (*a.k.a,*

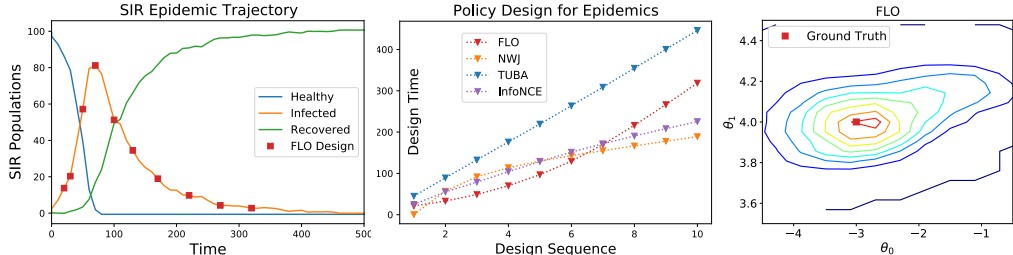

Figure 6: Diagnosis of learned sequential designs. The disease surveillance windows designed by FLO makes more sense: measures more frequently as infection spikes, and more sparsely when the pandemic slowly fades. The estimated parameter posterior (right) is consistent with the ground truth.

the *design parameters*), BOED tries to find the optimal data collection procedure that is expected to generate data that is most informative about the underlying model parameters, *i.e.*, solves for $\arg\max_d I(x; \theta; d)$. In this study, we focus on the more generic scenario where explicit likelihoods are not available, but we can still sample from the data generating procedure [43, 44].

We consider three carefully-selected models from recent literature for their progressive practical significance and the challenges involved [21, 41, 42]: static designs of ($i$) a simple linear regression model and ($ii$) a complex nonlinear pharmacokinetic model for drug development; and the dynamic policy design for ($iii$) epidemic disease surveillance and intervention (*e.g.*, for Covid-19 modeling). Designs with higher MI are more favorable, because it implies the data carries more information. In Figure 4 we compare design optimization curves using different MI optimization strategies, where FLO consistently leads. Popular NWJ and InfoNCE reports different tradeoffs that are less susceptible to FLO. We also examine the FLO predicted posteriors and confirm they are consistent with the ground-truth parameters (Figure 6 right). For the dynamic policy optimization, we also manually inspect the design strategies reported by different models (Figure 6 left,middle). Consistent with human judgement, FLO policy better assigns budgeted surveillance resources at different stages of pandemic progression.

**A novel meta-learning framework.** A second application of our work is to meta-learning, an area attracting substantial recent interest. In meta-learning, we are concerned with scenarios that at training time, there are abundant different labelled tasks, while upon deployment, only a handful of labeled instances are available to adapt the learner to a new task. Briefly, for an arbitrary loss $\ell_t(\hat{y}, y)$, where $t$ is the task identifier and $\hat{y} = f(x)$ is the prediction made by the model, we denote the risk by $R_t(f) = \mathbb{E}_{p_t(x,y)}[\ell_t(f(x), y)]$. Denote $R(f) \triangleq \mathbb{E}_{t \sim p(t)}[R_t(f)]$ as the expected risk for all tasks and $\hat{R}(f)$ for the mean of empirical risks computed from all training tasks. Inspired by recent information-theoretic generalization theories [84], we derived a novel, principled objective

$$\mathcal{L}_{\texttt{Meta-FLO}}(f) = \hat{R}(f) + \lambda\sqrt{I_{\texttt{FLO}}(\hat{\mathcal{D}}_t; \hat{E}_t)}, \tag{14}$$

where $\lambda$ is known given the data size and loss function, $(\hat{\mathcal{D}}_t, \hat{E}_t)$ are respectively data and task embeddings for training data, which for the first time lifts contrastive learning to the task and data distribution level. Our reasoning is that $\mathcal{L}_{\texttt{Meta-FLO}}(f)$ theoretically bounds $R(f)$ from above, and it is relatively sharp for being data-dependent. We give more information on this in the Appendix and defer a full exposition to a dedicated paper due to independent interest and space limits here. Note other MI bounds are not suitable for this task due to resource and vari-

Table 2: Multi-view representation learning on Cifar

| Model | InfoNCE | SpecNCE [34] [a] | FLO | FDV |
|---|---|---|---|---|
| MI | $5.73 \pm .07$ | $4.76 \pm .08$ | $5.83 \pm .08$ | $\mathbf{5.93 \pm .08}$ |

Figure 7: Few-shot adaptation with Meta-FLO.

[a]Note SpecNCE does not explicitly target mutual information

ance concerns. In Figure 7 we show Meta-FLO wins big over the state-of-the-art *model agnostic meta-learning* (MAML) model on the regression benchmark from [20].

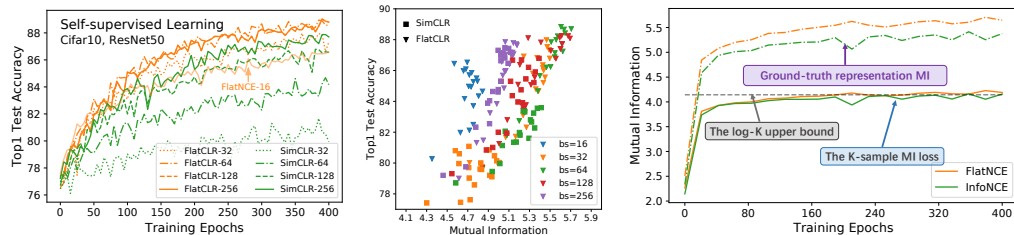

Figure 8: Sample efficiency comparison for `SimCLR` and `FlatCLR` on `Cifar10`.

Figure 9: Representation MI strongly correlates with performance.

Figure 10: `FlatNCE` better optimizes the true MI with the same mini-batch size. (`Cifar10` SSL training)

**Self-supervised learning (SSL)**. Finally, we wrap our experiments with one of the prime applications of contrastive MI estimation in machine learning: SSL for model pre-training[7]. Here we focus on how `FLO`-inspired objectives can improve the current practice of SSL, and given this topic's independent interest, we defer in-depth discussions in our dedicated work [12] where SSL-specific problems such as training diagnosis and low-precision numerical overflow are explored in detail. In this experiment, we follow the SSL setup described in the `SimCLR` paper [15]: in the pre-training phase, we optimize the mutual information between difference augmentations of the same image (*i.e.*, scaling, rotation, color jitting, *etc.*); and use linear probing accuracy as our performance criteria.

We compare the effectiveness of the `InfoNCE`-based `SimCLR` framework [15] to our `FLO`-based alternatives. To ensure fair comparison, we have used the FDV variant defined in Eq. (11) as our training objective, so that we are not introducing extra parameters to model $u(x, y)$. We call our new model `FlatCLR` because, perhaps counter-intuitively, the second term in Eq. (11) contributing all the learning signal is constant one in value (*i.e.*, being flat). In Figure 8 and 9, we show our new model `FlatCLR` shows superior sample efficiency compared to the SOTA `SimCLR` (a $8\times$ boost for the same performance, `FlatCLR`-32 versus `SimCLR`-256). This result is significant because `SimCLR`'s crucial reliance on large-batch training is a well-known limitation [34, 86, 46]. Figure 10 shows typical training curves with the respective models. Note that while the empirical estimates of MI are tied between the two methods, FDV optimized representation enjoys a better ground-truth MI [8], which can be explained by its robustness to the numerical overflow issue (see [12] for details). Further comparisons on the ground-truth MI estimation with different estimators can be found in Table 2.

## 4   Conclusion

We have described a new framework for the contrastive estimation of mutual information from energy modeling perspectives. Our work not only encapsulates popular variational MI bounds but also inspires novel objectives such as `FLO` and `FDV`, which comes with strong theoretical guarantees. In future work, we will leverage our theoretical insights to improve practical applications involving MI estimation, such as representation learning, fairness, causal inference [48], clustering [31], and in particular, data efficient learning. We are also interested in further exploring the deep connections between contrastive MI estimation and generative modeling [16, 73, 14, 18, 74, 13].

## Acknowledgements

The authors would like to thank the anonymous reviewers for their insightful comments. Q Guo gratefully appreciate the support of Amazon Fellowship. X Deng would like to thank the Advanced Research Computing program at Virginia Tech and Virginia's Commonwealth Cyber Initiative (CCI) AI testbed for providing computational resources, also appreciate the CCI and CCI-Coastal grants to Virginia Tech. Part of this work is done before C Tao joined Amazon, and he was funded by National Science Foundation Grant No. 1934964. This work used the Extreme Science and Engineering Discovery Environment (XSEDE), which is supported by National Science Foundation grant number ACI-1548562 [78] and used the Extreme Science and Engineering Discovery Environment (XSEDE) PSC Bridges-2 and SDSC Expanse at the service-provider through allocation TG-ELE200002 and TG-CIS210044.

---

[7]Code available in `https://github.com/Junya-Chen/FlatCLR`.

[8]Ground-truth MI is approximated by `InfoNCE` using a very large negative sample pool ($100\times$ mini-batch).

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
