# Tight Mutual Information Estimation With Contrastive Fenchel-Legendre Optimization

**Qing Guo**[1,†]**, Junya Chen**[2]**, Dong Wang**[2]**, Yuewei Wang**[2]**, Xinwei Deng**[1]
**Lawrence Carin**[2,3]**, Fan Li**[2]**, Jing Huang**[4]**, Chenyang Tao**[2,4,†]
[1]Virginia Tech [2]Duke University [3]KAUST [4]Amazon
qguo0701@vt.edu, chenyang.tao@duke.edu

# Appendix

## Table of Contents

36th Conference on Neural Information Processing Systems (NeurIPS 2022).

## A  Proof of Proposition 2.1 (InfoNCE Properties and derivation for some popular variational MI bounds)

*Proof.* Now let us prove `InfoNCE` is a lower bound to MI and under proper conditions this estimate is tight. Our proof is based on establishing that `InfoNCE` is a multi-sample extension of the `NWJ` bound. For completeness, we first repeat the proof for `BA` and `UBA` below, and then show `UBA` leads to `NWJ` and its multi-sample variant `InfoNCE`.

We can bound MI from below using an variational distribution $q(y|x)$ as follows:

$$
\begin{align}
I(X,Y) &= \mathbb{E}_{p(x,y)}\left[\log \frac{p(x,y)}{p(x)p(y)}\right] \tag{1}\\
&= \mathbb{E}_{p(x,y)}\left[\log \frac{p(y|x)p(x)q(y|x)}{p(x)p(y)q(y|x)}\right] \quad \text{\# q(y|x) is the variational distribution} \tag{2}\\
&= \mathbb{E}_{p(x,y)}\left[\log \frac{q(y|x)}{p(y)}\right] + \mathbb{E}_{p(x)}[\text{KL}(p(y|x)||q(y|x))] \tag{3}\\
&\geq \mathbb{E}_{p(x,y)}\left[\log q(y|x) - \log p(y)\right] \triangleq I_{\text{BA}}(X,Y;q) \tag{4}
\end{align}
$$

In sample-based estimation of MI, we do not know the ground-truth marginal density $p(y)$, which makes the above `BA` bound impractical. However, we can carefully choose an energy-based variational density that "cancels out" $p(y)$:

$$
q_f(y|x) = \frac{p(y)}{Z_f(x)}e^{f(x,y)}, \quad Z_f(x) \triangleq \mathbb{E}_{p(y)}[e^{f(x,y)}]. \tag{5}
$$

This auxiliary function $f(x,y)$ is known as the tilting function in importance weighting literature. Hereafter, we will refer to it the *critic function* in accordance with the nomenclature used in contrastive learning literature. The partition function $Z_f(x)$ normalizes this $q(y|x)$. Plugging this $q_f(y|x)$ into $I_{\text{BA}}$ yields:

$$
\begin{align}
I_{\text{BA}}(X,Y;q_f) &= \mathbb{E}_{p(x,y)}[f(x,y) + \log(p(y)) - \log(Z(x)) - \log p(y)] \tag{6}\\
&= \mathbb{E}_{p(x,y)}[f(x,y)] - \mathbb{E}_{p(x)}[\log(Z_f(x))] \triangleq I_{\text{UBA}}(X,Y;f) \tag{7}
\end{align}
$$

For $x, a > 0$, we have inequality $\log(x) \leq \frac{x}{a} + \log(a) - 1$. By setting $x \leftarrow Z(y)$ and $a \leftarrow e$, we have

$$
\log(Z(y)) \leq e^{-1}Z(y). \tag{8}
$$

Plugging this result into (7) we recover the celebrated `NWJ` bound, which lower bounds $I_{\text{UBA}}$:

$$
I_{\text{UBA}}(X,Y) \geq \mathbb{E}_{p(x,y)}[f(x,y)] - e^{-1}\mathbb{E}_{p(x)}[Z_f(x)] \triangleq I_{\text{NWJ}}(X,Y;f). \tag{9}
$$

When $f(x,y)$ takes the value of

$$
f^*(x,y) = 1 + \log \frac{p(x|y)}{p(x)}, \tag{10}
$$

this bound is sharp.

We next extend these bounds to the multi-sample setting. In this setup, we are given one paired sample $(x_1, y_1)$ from $p(x,y)$ (*i.e.*, the positive sample) and $K - 1$ samples independently drawn from $p(y)$ (*i.e.*, the negative samples). Note that when we average over $x$ wrt $p(x)$ to compute the MI, this equivalent to comparing positive pairs from $p(x,y)$ and negative pairs artificially constructed by $p(x)p(y)$. By the independence between $X_1$ and $Y_{k>1}$, we have

$$I(X; Y_{1:K}) = \mathbb{E}_{p(x_1, y_1) \prod_{k>1} p(y_k)} \left[ \frac{p(x_1, y_1) \prod_{k>1} p(y_k)}{p(x_1) \prod_k p(y_k)} \right] = \mathbb{E}_{p(x_1, y_1)} \left[ \frac{p(x_1, y_1)}{p(x_1) p(y_1)} \right] = I(X; Y) \tag{11}$$

So for arbitrary multi-sample critic $f(x; y_{1:K})$, we know

$$I(X; Y) = I(X_1; Y_{1:K}) \geq I_{\texttt{NWJ}}(X_1, Y_{1:K}; f) = \mathbb{E}_{p(x_1, y_1) \prod_{k>1} p(y_k)}[f(x_1, y_{1:K})] - e^{-1} \mathbb{E}_{p(x)}[Z_f(x)] \tag{12}$$

Now let us set

$$\tilde{f}(x_1; y_{1:K}) = 1 + \log \frac{e^{g(x_1, y_1)}}{m(x_1; y_{1:K})}, \quad m(x_1; y_{1:K}) = \frac{1}{K} \sum_k e^{g(x_1, y_k)}. \tag{13}$$

$$I_{\texttt{NWJ}}(X_1, Y_{1:K}; \tilde{f}) = \mathbb{E}_{p(x_1, y_1) p^{K-1}(y_k)} \left[ 1 + \log \frac{e^{g(x_1, y_1)}}{m(x_1; y_{1:K})} \right] - \mathbb{E}_{p(x') p^K(y')} \left[ e^{-1 + 1 + \log \frac{e^{g(x_1', y_1')}}{m(x_1'; y_{1:K}')}} \right]$$

$$= \mathbb{E}_{p(x_1, y_1) p^{K-1}(y_k)} \left[ 1 + \log \frac{e^{g(x_1, y_1)}}{m(x_1; y_{1:K})} \right] - \mathbb{E}_{p(x') p^K(y')} \left[ \frac{e^{g(x_1', y_1')}}{m(x_1'; y_{1:K}')} \right]$$

Due to the symmetry of $\{y_k\}_{k=1}^K$, we have

$$\mathbb{E}_{p(x') p^K(y')} \left[ \frac{e^{g(x_1', y_1')}}{m(x_1'; y_{1:K}')} \right] = \mathbb{E}_{p(x') p^K(y')} \left[ \frac{e^{g(x_1', y_k')}}{m(x_1'; y_{1:K}')} \right]. \tag{14}$$

So this gives

$$\mathbb{E}_{p(x') p^K(y')} \left[ \frac{e^{g(x_1', y_1')}}{m(x_1'; y_{1:K}')} \right] = \mathbb{E}_{p(x') p^K(y')} \left[ \frac{\frac{1}{K} e^{g(x_1', y_k')}}{m(x_1'; y_{1:K}')} \right] = 1, \tag{15}$$

and one can easily see this recovers the $K$-sample `InfoNCE` defined in (3)

$$I_{\texttt{NWJ}}(X_1, Y_{1:K}; \tilde{f}) = \mathbb{E}_{p(x_1, y_1) p^{K-1}(y_k)} \left[ \log \frac{e^{g(x_1, y_1)}}{m(x_1; y_{1:K})} \right] = I_{\texttt{InfoNCE}}^K(X; Y | g) \tag{16}$$

Now we need to show this bound is sharp when $K \to \infty$. We only need to show that for some choice of $g(x, y)$, the inequality holds asymptotically. Recall the `NWJ`'s optimal critic takes value of $f^*(x, y) = 1 + \frac{p(x|y)}{p(x)}$, so with reference to (13) let us plug in $g^*(x, y) = \frac{p(y|x)}{p(y)}$ into `InfoNCE`

$$\mathcal{L}_K^* = \mathbb{E}_{p^K} \left[ \log \left( \frac{f^*(x_k, y_k)}{f^*(x_k, y_k) + \sum_{k' \neq k} f^*(x_k, y_{k'})} \right) \right] + \log K \tag{17}$$

$$= -\mathbb{E} \left[ \log \left( 1 + \frac{p(y)}{p(y|x)} \sum_{k'} \frac{p(y_{k'}|x_k)}{p(y_{k'})} \right) \right] + \log K \tag{18}$$

$$\approx -\mathbb{E} \left[ \log \left( 1 + \frac{p(y)}{p(y|x)} (K-1) \mathbb{E}_{y_{k'}} \frac{p(y_{k'}|x_k)}{p(y_{k'})} \right) \right] + \log K \tag{19}$$

$$= -\mathbb{E} \left[ \log \left( 1 + \frac{p(y_k)}{p(y_k|x_k)} (K-1) \right) \right] + \log K \tag{20}$$

$$\approx -\mathbb{E} \left[ \log \frac{p(y)}{p(y|x)} \right] - \log(K-1) + \log K \tag{21}$$

$$(K \to \infty) \quad \to \quad I(X; Y) \tag{22}$$

This concludes our proof. $\qquad \square$

## B  Proof of Proposition 2.2 (FLO lower bounds MI)

*Proof.* The proof is given in line 133-140 in the main text. Basically we have applied the Fenchel duality trick to the $\log$ term in the UBA bound. Note that unlike UBA, our FLO bound can be unbiased estimated with finite samples (as UBA requires an infinite sum inside its $\log$ term, which makes finite-sample empirical estimate biased per Jensen's inequality). ☐

## C  Proof of Proposition 2.3, Corollary 2.4 (FLO tightness, meaning of $u(x, y)$)

*Proof.* The proof is given in the main text, more specifically the paragraph preceding Proposition 2.3. ☐

## D  Gradient Analysis of FLO (More Detailed)

To further understand the workings of FLO, let us inspect the gradient of model parameters. Recall the intractable UBA MI estimator can be re-expressed in the following form:

$$I_{\text{UBA}'}(g_\theta) = \mathbb{E}_{p(x,y)}[-\log \mathbb{E}_{p(y')}[\exp(g_\theta(x, y') - g_\theta(x, y))]]] \tag{23}$$

In this part, we want to establish the intuition that $\nabla_\theta\{I_{\text{FLO}}(u_\phi, g_\theta)\} \approx \nabla_\theta\{I_{\text{UBA}'}(g_\theta)\}$, where

$$I_{\text{FLO}}(u_\phi, g_\theta) \triangleq -\left\{ u_\phi(x, y) + \mathbb{E}_{p(y')}[\exp(-u_\phi(x, y) + g_\theta(x, y') - g_\theta(x, y))] \right\} \tag{24}$$

is our FLO estimator.

By defining

$$\mathcal{E}_\theta(x, y) \triangleq \frac{1}{\mathbb{E}_{p(y')}[\exp(g_\theta(x, y') - g_\theta(x, y))]}, \tag{25}$$

we have

$$\nabla_\theta\left\{\frac{1}{\mathcal{E}_\theta(x, y)}\right\} = -\frac{\nabla\mathcal{E}_\theta(x, y)}{(\mathcal{E}_\theta(x, y))^2} = -\frac{\nabla_\theta \log \mathcal{E}_\theta(x, y)}{\mathcal{E}_\theta(x, y)}, \tag{26}$$

and

$$\nabla_\theta\left\{\frac{1}{\mathcal{E}_\theta(x, y)}\right\} = \nabla_\theta \mathbb{E}_{p(y')}[\{\exp(g_\theta(x, y') - g_\theta(x, y))\}] \tag{27}$$

$$= \mathbb{E}_{p(y')}[\nabla_\theta\{\exp(g_\theta(x, y') - g_\theta(x, y))\}]. \tag{28}$$

We know fixing $g_\theta(x, y)$, the corresponding optimal $u_\theta^*(x, y)$ maximizing FLO is given by

$$u_\theta^*(x, y) = \log \mathbb{E}_{p(y')}[\exp(g_\theta(x, y') - g_\theta(x, y))] = -\log \mathcal{E}_\theta(x, y). \tag{29}$$

This relation implies the view that $\exp^{-u_\phi(x,y)}$ is optimized to approximate $\mathcal{E}_\theta(x, y)$. And to emphasize this point, we now write $\hat{\mathcal{E}}_\theta(x, y) \triangleq e^{-u_\phi(x,y)}$. Assuming this approximation is sufficiently accurate (*i.e.*, $\mathcal{E}_\theta \approx \hat{\mathcal{E}}_\theta$), we have

$$\nabla_\theta\{I_{\text{FLO}}(u_\phi, g_\theta)\} = -\mathbb{E}_{p(x,y)}\left[e^{-u_\phi(x,y)}\mathbb{E}_{p(y')}[\nabla_\theta \exp(g_\theta(x, y') - g_\theta(x, y))]\right] \tag{30}$$

$$= \mathbb{E}_{p(x,y)}\left[\frac{e^{-u_\phi(x,y)}}{\mathcal{E}_\theta(x, y)}\nabla_\theta \log \mathcal{E}_\theta(x, y)\right] \tag{31}$$

$$= \mathbb{E}_{p(x,y)}\left[\frac{\hat{\mathcal{E}}_\theta(x, y)}{\mathcal{E}_\theta(x, y)}\nabla_\theta \log \mathcal{E}_\theta(x, y)\right] \tag{32}$$

$$\approx \mathbb{E}_{p(x,y)}[\nabla_\theta \log \mathcal{E}_\theta(x, y)] \tag{33}$$

$$= \nabla_\theta\{\mathbb{E}_{p(x,y)}[\log \mathcal{E}_\theta(x, y)]\} = \nabla_\theta\{I_{\text{UBA}'}(g_\theta)\}. \tag{34}$$

While the above relation shows we can use FLO to amortize the learning of UBA, one major caveat with the above formulation is that $\hat{u}(x, y)$ has to be very accurate for it to be valid. As such, one needs to solve a cumbersome nested optimization problem: update $g_\theta$, then optimize $u_\phi$ until it converges before the next update of $g_\theta$. Fortunately, we can show that is unnecessary: the convergence can be established under much weaker conditions, which justifies the use of simple simultaneous stochastic gradient descent for both $(\theta, \phi)$ in the optimization of FLO.

# E Proof of Proposition 2.5 (FLO Convergence under SGD)

Our proof is based on the convergence analyses of generalized stochastic gradient descent from [20]. We cite the main assumptions and results below for completeness.

**Definition E.1** (Generalized SGD, Problem 2.1 in [20]). Let $h(\theta; \omega), \omega \sim p(\omega)$ be an unbiased stochastic gradient estimator for objective $f(\theta)$, $\{\eta_t > 0\}$ is the fixed learning rate schedule, $\{\xi_t > 0\}$ is the random perturbations to the learning rate. We want to solve for $\nabla f(\theta) = 0$ with the iterative scheme $\theta_{t+1} = \theta_t + \tilde{\eta}_t h(\theta_t; \omega_t)$, where $\{\omega_t\}$ are iid draws and $\tilde{\eta}_t = \eta_t \xi_t$ is the randomized learning rate.

**Assumption E.2.** (Standard regularity conditions for Robbins-Monro stochastic approximation, Assumption D.1 [20]).

  A1. $h(\theta) \triangleq \mathbb{E}_\omega[h(\theta; \omega)]$ is Lipschitz continuous;

  A2. The ODE $\dot{\theta} = h(\theta)$ has a unique equilibrium point $\theta^*$, which is globally asymptotically stable;

  A3. The sequence $\{\theta_t\}$ is bounded with probability 1;

  A4. The noise sequence $\{\omega_t\}$ is a martingale difference sequence;

  A5. For some finite constants $A$ and $B$ and some norm $\|\cdot\|$ on $\mathbb{R}^d$, $\mathbb{E}[\|\omega_t\|^2] \leq A + B\|\theta_t\|^2$ a.s. $\forall t \geq 1$.

**Proposition E.3** (Generalized stochastic approximation, Proposition 2.2 in [20])**.** Under the standard regularity conditions listed in Assumption E.2, we further assume $\sum_t \mathbb{E}[\tilde{\eta}_t] = \infty$ and $\sum_t \mathbb{E}[\tilde{\eta}_t^2] < \infty$. Then $\theta_n \to \theta^*$ with probability 1 from any initial point $\theta_0$.

**Assumption E.4.** (Weaker regularity conditions for generalized Robbins-Monro stochastic approximation, Assumption G.1 in [20]).

  B1. The objective function $f(\theta)$ is second-order differentiable.

  B2. The objective function $f(\theta)$ has a Lipschitz-continuous gradient, i.e., there exists a constant $L$ satisfying

$$-LI \preceq \nabla^2 f(\theta) \preceq LI,$$

  B3. The noise has a bounded variance, i.e., there exists a constant $\sigma > 0$ satisfying $\mathbb{E}\left[\|h(\theta_t; \omega_t) - \nabla f(\theta_t)\|^2\right] \leq \sigma^2$.

**Proposition E.5** (Weaker convergence results, Proposition G.2 in [20])**.** Under the technical conditions listed in Assumption E.4, the SGD solution $\{\theta_t\}_{t>0}$ updated with generalized Robbins-Monro sequence ($\tilde{\eta}_t$: $\sum_t \mathbb{E}[\tilde{\eta}_t] = \infty$ and $\sum_t \mathbb{E}[\tilde{\eta}_t^2] < \infty$) converges to a stationary point of $f(\theta)$ with probability 1 (equivalently, $\mathbb{E}\left[\|\nabla f(\theta_t)\|^2\right] \to 0$ as $t \to \infty$).

*Proof.* Since $\hat{\mathcal{E}}_{\theta_t}/\mathcal{E}_{\theta_t}$ is bounded between $[a, b]$ ($0 < a < b < \infty$), results follow by a direct application of Proposition E.3 and Proposition E.5. $\square$

# F Gaussian Toy Model Experiments

First, we start validating the properties and utility of the proposed FLO estimator by comparing it to competing solutions with the Gaussian toy models. Specifically, for the $2d$-D Gaussian model with correlation $\rho$, we have $X \in \mathbb{R}^d$ and $Y \in \mathbb{R}^d$ with covariance structure

$$\text{cov}[[X]_i, [X]_j] = \delta_{ij}, \text{cov}[[Y]_i, [Y]_j] = \delta_{ij}, \text{cov}[[X]_i, [Y]_j] = \delta_{ij} \cdot \rho \qquad (35)$$

This allows us to have the ground-truth MI $I(X; Y) = -\frac{d}{2}\log(1 - \rho^2)$ for reference and easily tune the difficulty of the task via varying $d$ and $\rho$.

## F.1 Choice of baselines

We choose TUBA, NWJ, InfoNCE and $\alpha$-InfoNCE as our baselines. Note $\alpha$-InfoNCE results are not reported in the main paper because we do not see a clear advantage via tuning $\alpha$ NWJ and InfoNCE

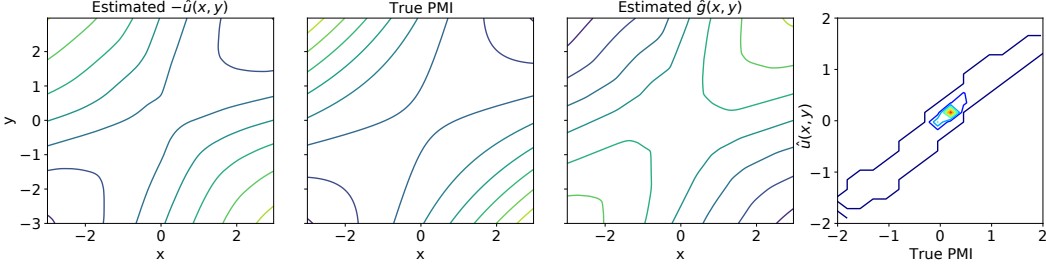

Figure S1: Comparison of estimated $u(x, y)$, $g(x, y)$ and the ground-truth PMI $-\log \frac{p(x,y)}{p(x)p(y)}$ using the 2D Gaussian experiment. This confirms our analyses that the optimized $u(x, y)$ approximates the true PMI.

are the two most popular estimators in practice that are employed without additional hacks. TUBA is included for its close relevance to FLO (*i.e.*, optimizing $u(x)$ instead of $u(x, y)$, and being non-contrastive). We do not include DV here because we find DV needs excessively a large negative sample size $K$ to work. Variants like MINE are excluded for involving additional tuning parameters or hacks which complicates our analyses. The proposed FDV estimator is also excluded from our analyses for bound comparison since it includes $\hat{I}_{\text{DV}}$ in the estimator. Note that although not suitable for MI estimation, we find FDV works quite well in representation learning settings where the optimization of MI is targeted. This is because in FDV, the primal term $\hat{I}_{\text{DV}}$ term does not participate gradient computation, so it does not yield degenerated performance as that of DV. In the results reported below, we fixed $\alpha = 0.8$ for better visualization.

## F.2 Experimental setups

We use the following baseline setup for all models unless otherwise specified. For the critic functions $g(x, y)$, $u(x, y)$ and $u(x)$, we use multi-layer perceptron (MLP) network construction with hidden-layers $512 \times 512$ and ReLU activation. For optimizer, we use Adam and set learning rate to $10^{-4}$ unless otherwise sepcified. A default batch-size of $128$ is used for training. To report the estimated MI, we use $10k$ samples and take the average. To visualize variance, we plot the decimal quantiles at $\{10\%, 20\%, \cdots, 80\%, 90\%\}$ and color code with different shades. We sample fresh data point in each iteration to avoid overfitting the data. All models are trained for $\sim 5{,}000$ iterations (each epoch samples $10k$ new data points, that is $78$ iterations per epoch for a total of $50$ epochs).

## F.3 PMI approximation with $u(x, y)$

For Figure S1, we use the 2-D Gaussian with $\rho = 0.5$ to compare the estimated $u(x, y)$, $g(x, y)$ with the ground-truth PMI, and the contour plot is obtained with a grid resolution of $2.5 \times 10^{-2}$. This confirms our analyses that the optimized $u(x, y)$ approximates the true PMI $-\log \frac{p(x,y)}{p(x)p(y)}$.

## F.4 Ablation study: efficiency of parameter sharing for $g(x, y)$ and $u(x, y)$.

For the shared parameterization experiment for FLO (Figure S2), we used the more challenging 20-D Gaussian with $\rho = 0.5$, and trained the network with learning rate $10^{-3}$ and $10^{-4}$ respectively. We repeat the experiments for $10$ times and plot the distribution of the MI estimation trajectories. Note that we intentionally used a setup such that the MLP network architecture we used is inadequate to get a sharp estimate (both for FLO and other MI estimators), which simulates the realistic scenario that the ground-truth MI is infeasible due to architecture constraints (refer to our ablation study on the influence network capacity in Sec F.5). We observe the FLO estimator with a shared network learns faster than its separate network counterpart under both learning rates, validating the superior efficiency of parameter sharing.

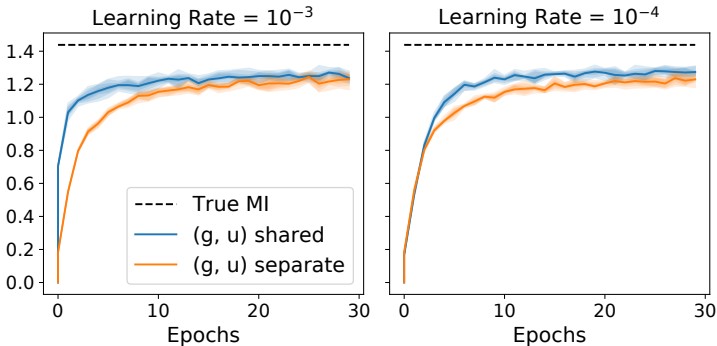

Figure S2: MI estimation with different critic parameter sharing strategies for FLO: shared network and separate networks under learning rates $10^{-3}$ and $10^{-4}$ for 2-D Gaussian. Note shared parameterization not only reduced half the network size, it also learns faster.

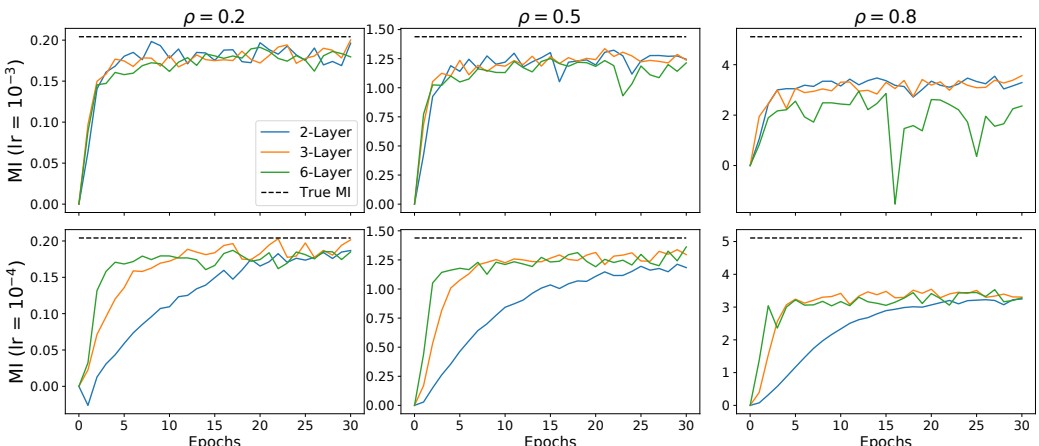

Figure S3: Abaltion study for network complexity with FLO. More complex networks lead to faster convergence and better MI estimates. However, the stability is more sensitive to learning rate with a larger neural network.

### F.5 Ablation study: network capacity and MI estimation accuracy

We further investigate how the neural network learning capacity affect MI estimation. In Figure S3 we compare the training dynamics of the FLO estimator with $L$-layer neural networks, where $L \in \{2, 3, 6\}$ and each hidden-layer has $512$-units. A deeper network is generally considered to be more expressive. We see that using larger networks in general converge faster in terms of training iterations, and also obtain better MI estimates. However, more complex networks imply more computation per iteration, and it can be less stable when trained with larger learning rates.

### F.6 Ablation study: Bi-linear critics and scaling

We setup the *bi-linear* critic experiment as follows. For the naive baseline FLO, we use the shared-network architecture for $g(x, y)$ and $u(x, y)$, and use the in-batch shuffling to create the desired number of negative samples (FLO-shuff). For FLO-BiL, we adopt the following implementation: feature encoders $h(x), \tilde{h}(y)$ are respectively modeled with three layer MLP with $512$-unit hidden layers and ReLU activations, and we set the output dimension to $512$. Then we concatenate the feature representation to $z = [h(x), \tilde{h}(y)]$ and fed it to the $u(x, y)$ network, which is a two-layer $128$-unit MLP. Note that is merely a convenient modeling choice and can be further optimized for efficiency. Each epoch containing $10k$ samples, and FLO-shuff is trained with fixed batch-size. For FLO-BiL, it is trained with batch-size set to the negative sample-size desired, because all in-batch data are served as negatives. We use the same learning rate $10^{-4}$ for both cases, and this puts large-batch training at disadvantage, as fewer iterations are executed. To compensate for this, we use $T(K) = (\frac{K}{K_0})^{\frac{1}{2}} \cdot T_0$ to set the total number of iterations for FLO-BiL, where $(T_0, K_0)$ are respectively the baseline

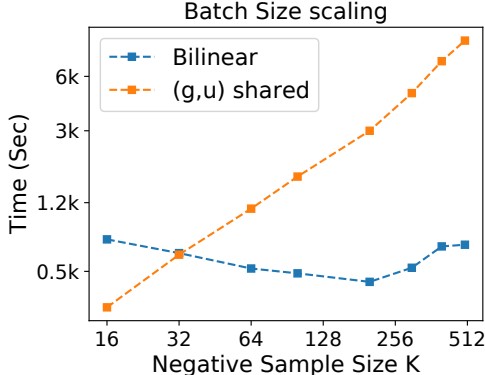

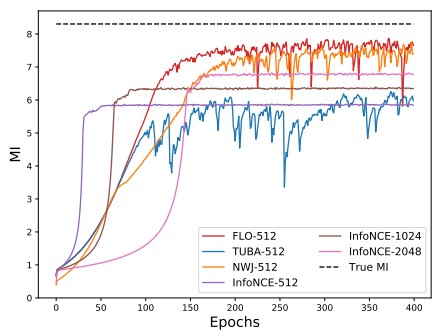

Figure S4: Comparison of computation time of the shared MLP critic and the bi-linear critic. Overall the bilinear implementation is more efficient than the shared MLP. `FLO`'s initial drop in computation time with growing negative sample size is due to better exploitation of parallel computation.

Figure S5: Comparison of learning dynamics with 20-D Gaussian at $\rho = 0.9$. We used bi-linear critics for all bounds. Note `InfoNCE` enjoys stable learning, and its convergence is fast in the small-sample regime but slow in the large-sample regime. In all cases `InfoNCE` suffers form large biases. `NWJ` is more accurate but it learns slower. In contrast, our `FLO` learns fast and stably.

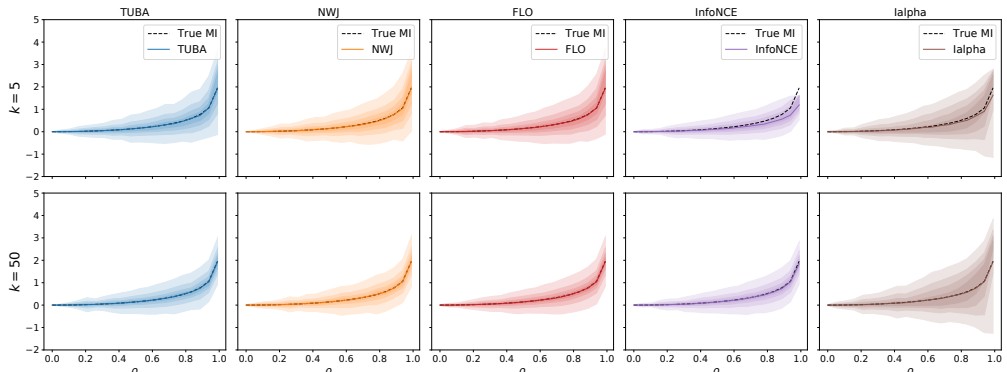

Figure S6: Bias variance plot for the popular MI bounds with the 2-D Gaussians. In this simpler case, `TUBA`, `NWJ` and `FLO` all give sharp estimate at $K = 5$. $\alpha$-`InfoNCE` gives worst variance profile. The reason is that because $\alpha$-`InfoNCE` interpolates between the low-variance multi-sample `InfoNCE` and high-variance single-sample `NWJ` (see Figure S7), and in this case the variance from `NWJ` dominates.

training iteration and negative sample size used by `FLO-shuff`, and the number of negative sample K are $\{10, 50, 100, 150, 200, 250, 300, 350, 400, 450, 500\}$. We are mostly interested in computation efficiency here so we do not compare the bound. In Figure S4, we see the cost for training `FLO-shuff` grows linearly as expected. For `FLO-BiL`, a U-shape cost curve is observed. This is because bilinear implementation has three networks total, while the shared MLP only have one network. This implies more computations when the batch size is small, however, as the batch size grows, the computation overhead is amortized by better parallelism employed with the bilinear strategy, thus increasing overall efficiency until the device capacity has been reached. This explains the initial drop in cost, followed by the anticipated square-root growth.

## F.7    Comparison of learning dynamics for different variational MI bounds

In Figure S5, we show the learning dynamics of competing estimators for the 20-D Gaussian when $\rho = 0.9$. We can find `FLO` achieves the best accuracy, it also learns fast and stably. `InfoNCE` learns very stably, yet its learning efficiency varies significantly in small-batch and large-batch setups.

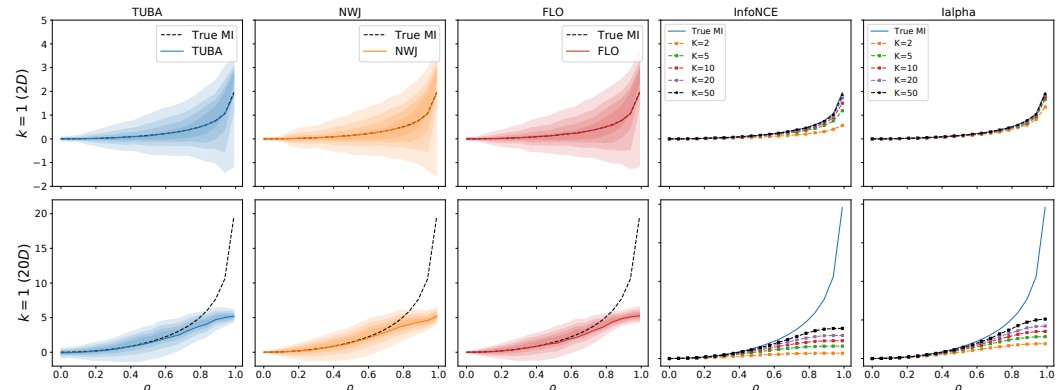

Figure S7: Bias variance plot for the popular MI bounds with the 2-D (upper panel) and 20-D (lower panel) Gaussians. Single-sample estimator of TUBA, NWJ and FLO (*i.e.*, $K = 1$) are compared to the multi-sample estimators of InfoNCE and $\alpha$-InfoNCE.

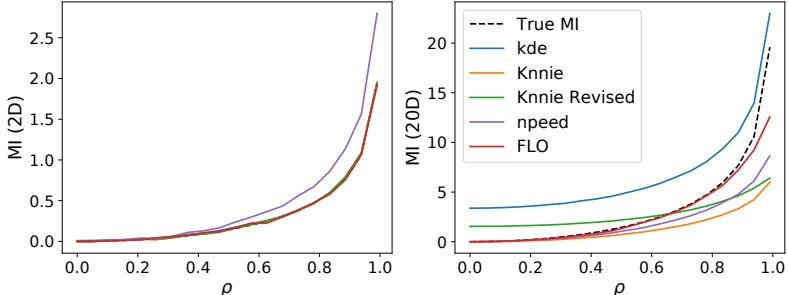

Figure S8: Comparison to classical MI estimators. (left) Easy 2D Gaussian, all models perform similarly. (right) Challenging 20D Gaussian, where FLO shows better overall accuracy. Note that the kde accuracy in the high-dimensional setting is mis-judged, as it is well-known kernel-based density estimator scale poorly in high-dimensions.

## F.8 Comprehensive analyses of bias-variance trade-offs

To supplement our results in the main paper, here we provide additional bias-variance plots for different MI estimators under various settings. In Figure S6 we show the bias-variance plot of MI estimates for 2-D Gaussians. In this case, the network used are sufficiently comprehensive so sharp estimate is attainable. In all cases the estimation variance grows with MI value, which is consistent with the theoretical prediction that for tight estimators, the estimation variance grows exponential with MI [17]. In such cases, the argument for InfoNCE's low-variance profile no longer holds: it is actually performing sub-optimally. For complex real applications, the negative sample size used might not provide an adequate estimate of ground-truth MI (*i.e.*, the $\log K$ cap), and that is when InfoNCE's low-variance profile actually helps. We also notice that, when the MI estimate is not exactly tight, but very close to the true value, the variance dropped considerably. This might provide alternative explanation (and opportunity) for the development near-optimal MI estimation theories, which is not covered in existing literature.

We also tried the single-sample estimators for NWJ, TUBA and FLO to their multi-sample InfoNCE-based counterparts (Figure S7), which is the comparison made by some of the prior studies (Note we do not apply Bilinear tric here, thus FLO seems similar to other methods). In this setting, the variance single-sample estimators' variances are considerably larger, which explains their less favorable performance. Note that contradictory to theoretical predictions, a larger negative sample size does make NWJ, TUBA and FLO tighter empirically, although the gains are much lesser compare to that of InfoNCE (partly because these three estimators are already fairly tight relative to InfoNCE). This might be explained by a better optimization landscape due to reduced estimation variance. We conjecture that for multi-sample NWJ, TUBA and FLO, the performance in empirical applications such as self-supervised learning should be competitive to that of InfoNCE, which has never been reported in literature.

## G Comparison with Classical MI Estimators

We also compare our FLO estimator to the classical MI estimators in Figure S8. The following implementations of baseline estimators for multi-dimensional data are considered: (*i*) *KDE*: we use kernel density estimators to approximate the joint and marginal likelihoods, then compute MI by definition; (*ii*) *NPEET* [1], a variant of Kraskov's $K$-nearest neighbour (KNN) estimator [15, 21]; (*iii*) *KNNIE* [2], the original KNN-estimator and its revised variant [10]. These models are tested on 2-D and 20-D Gaussians with varying strength of correlation, with their hyper-parameters tuned for best performance. Note that the notation of "best fit" is a little bit subjective, as we will fix the hyper-parameter for all dependency strength, and what works better for weak dependency might necessarily not work well for strong dependency. We choose the parameter whose result is visually most compelling. In addition to the above, we have also considered other estimators such as maximal-likelihood density ratio [3] [19] and KNN with local non-uniformity correction [4]. However, these models either do not have a publicly available multi-dimensional implementation, or their codes do not produce reasonable results [5].

## H Comparison to Parametric Variational Estimators and Bounds Targeting Alternative Information Metrics

Parametric variational estimators are typically associated with upper bound of MI [8, 18]. Inspired by multi-sample variational bounds for likelihood estimation, [7] derived a generic family of importance-weighted MI bounds that are provably tighter. These bounds usually require the additional knowledge of likelihood, and consequently they can not be directly used for data-driven MI estimations. On the other hand, these models do not suffer from the exponential scaling of variance suffered by non-parametric MI estimators. Note that MI is not the only measure to assess association between two random variables, some alternatives can potentially do better for specific applications. Examples include $\mathcal{V}$ information [22], Rényi information [16], and the spectral information [11].

## I Regression with Sensitive Attributes (Fair Learning) Experiments

### I.1 Introduction to fair machine learning

Nowadays consequential decisions impacting people's lives have been increasingly made by machine learning models. Such examples include loan approval, school admission, and advertising campaign, amongst others. While automated decision making has greatly simplified our lives, concerns have been raised on (inadvertently) echoing, even amplifying societal biases. Specially, algorithms are vulnerable in inheriting discrimination from the training data and passed on such prejudices in their predictions.

To address the growing need for mitigating algorithmic biases, research has been devoted in this direction under the name fair machine learning. While discrimination can take many definitions that are not necessarily compatible, in this study we focus on the most widely recognized criteria *Demographic Parity* (DP), as defined below

**Definition I.1** (Demographic Parity, [9])**.** The absolute difference between the selection rates of a decision rule $\hat{y}$ of two demographic groups defined by sensitive attribute $s$, *i.e.*,

$$\mathrm{DP}(\hat{Y}, S) = \left| \mathbb{P}(\hat{Y} = 1|S = 1) - \mathbb{P}(\hat{Y} = 1|S = 0) \right|. \tag{36}$$

With multiple demographic groups, it is the maximal disparities between any two groups:

$$\mathrm{DP}(\hat{Y}, S) = \max_{s \neq s'} \left| \mathbb{P}(\hat{Y} = 1|S = s) - \mathbb{P}(\hat{Y}|S = s') \right|. \tag{37}$$

---

[1] https://github.com/gregversteeg/NPEET
[2] https://github.com/wgao9/knnie
[3] https://github.com/leomuckley/maximum-likelihood-mutual-information
[4] https://github.com/BiuBiuBiLL/NPEET_LNC
[5] These are third-party python implementations, so BUGs are highly likely.

## I.2 Experiment details and analyses

To scrub the sensitive information from data, we consider the *in-processing* setup

$$\mathcal{L} = \text{Loss}(\underbrace{\text{Predictor}(\text{Encoder}(x_i)), y_i}_{\text{Primary loss}}) + \lambda \underbrace{I(s_i, \text{Encoder}(x_i))}_{\text{Debiasing}}. \tag{38}$$

By regularizing model training with the violation of specified fairness metric $\Delta(\hat{y}, s)$, fairness is enforced during model training. In practice, people recognize that appealing to fairness sometimes cost the utility of an algorithm (*e.g.*, prediction accuracy) [12]. So most applications seek to find their own sweet points on the fairness-utility curve. In our example, it is the *DP-error* curve. A fair-learning algorithm is consider good if it has lower error at the same level of DP control.

In this experiment, we compare our MI-based fair learning solutions to the state-of-the-art methods. *Adversarial debiasing* tries to maximize the prediction accuracy for while minimize the prediction accuracy for sensitivity group ID [23]. We use the implementation from `AIF360`[6] package [6]. FERMI is a density-based estimator for the *exponential Rényi mutual information* $\text{ERMI} \triangleq \mathbb{E}_{p(x,y)}[\frac{p(x,y)}{p(x)p(y)}]$, and we use the official codebase. For evaluation, we consider the *adult* data set from UCI data repository [5], which is the 1994 census data with 30k samples in the train set and 15k samples in the test set. The target task is to predict whether the income exceeds \$50k, where gender is used as protected attribute. Note that we use this binary sensitive attribute data just to demonstrate our solution is competitive to existing solutions, where mostly developed for binary sensitive groups. Our solution can extend to more general settings where the sensitive attribute is continuous and high-dimensional.

We implement our fair regression model as follows. To embrace data uncertainty, we consider latent variable model $p_\theta(y, x, z) = p_\theta(y|z)p_\theta(x|z)p(z)$, where $v = \{x, y\}$ are the observed predictor and labels. Under the variational inference framework [13], we write the $\text{ELBO}(v; p_\theta(v, z), q_\phi(z|v))$ as

$$\mathbb{E}_{Z \sim q_\phi(z|v)}[\log p_\theta(y|Z)] + \mathbb{E}_{Z \sim q_\phi(z|v)}[\log p_\theta(x|Z)] - \beta \text{KL}(q_\phi(z|v) \parallel p(z)) \tag{39}$$

$p(z)$ is modeled with standard Gaussian, and the approximate posterior $q_\phi(z|v)$ is modeled by a neural network parameterizing the mean and variance of the latents (we use the standard mean-field approximation so cross-covariance is set to zero), and $\beta$ is a hyperparameter controlling the relative contribution of the KL term to the objective. Note that unlike in the standard ELBO we have dropped the term $\mathbb{E}_{Z \sim q_\phi(z|v)}[\log p_\theta(x|Z)]$ because we are not interested in modeling the covariates. Note this coincides with the *variational information bottleneck* (VIB) formulation [1]. Additionally, the posterior $q_\phi(z|v)$ will not be conditioned on $y$, but only on $x$, because in practice, the labels $y$ are not available at inference time. All networks used here are standard three-layer MLP with 512 hidden-units.

For Figure S9, we note that the adversarial de-biasing actually crashed in the DP range $[0.1, 0.18]$, so the results have to be removed. Since interpolation is used to connect different data points, it makes the adversarial scheme look good in this DP range, which is not the case. FERMI also gave unstable estimation in the DP range $[0.1, 0.18]$. Among the MI-based solutions, `NWJ` was most unstable. Performance-wise, `InfoNCE`, `TUBA` and `FDV` are mostly tied, with the latter two slightly better in the "more fair" solutions (*i.e.*, at the low DP end).

# J  Self-supervised Learning

Our codebase is modified from a public `PyTorch` implementation[7]. Specifically, we train 256-dimensional feature representations by maximizing the self-MI between two random views of data, and report the test set classification accuracy using a linear classifier trained to convergence. We report performance based on `ResNet-50`. Hyper-parameters are adapted from the original `SimCLR` paper. For the large-batch scaling experiment, we first grid-search the best learning rate for the base batch-size, then grow the learning rate linearly with batch-size.

---

[6] https://github.com/Trusted-AI/AIF360
[7] https://github.com/sthalles/SimCLR

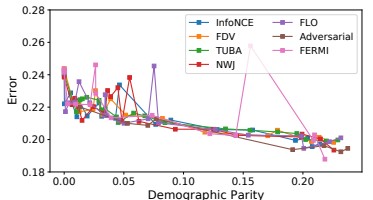

Figure S9: Fair Learning Result.

Table S1: MNIST cross-view results.

| Model | CCA | NWJ | TUBA | InfoNCE | FLO | FDV |
|---|---|---|---|---|---|---|
| Accuracy | 67.78 | 76.71 | 79.49 | 79.27 | 79.47 | **80.14** |
| $\hat{I}(x_l, x_r)$ | NA | 5.73 | 4.78 | 4.65 | 4.84 | 4.67 |

# K    Bayesian Experimental Design

## K.1    Noisy Linear Model

Our setup is the same as the Noisy Linear Model in [14]. We use 10 individual experimental designs. For encoder $\theta$ and encoder $y$, we use MLP with 2-layer, 128-dim hidden layer, and set the feature dim as 512. We train models in 5000 epochs, the batch size is 64, and the learning rate is $2 * 10^{-5}$. Four MI estimators (NWJ, TUBA, InfoNCE, and FLO) has been compared in this experiment and we got four optimized designs. Then, we use MCMC to estimate the posterior of the parameters.

## K.2    Pharmacokinetic Model

The settings of this experiment refer to the Pharmacokinetic Model of [14]. We use 10 individual experimental designs. The MLP is with 2-layer, 128-dim hidden layer, and set the output feature dim as 512. We train 10000 epochs with learning rate is $10^{-5}$ via four methods (NWJ, TUBA, InfoNCE, FLO).

## K.3    SIR Model

We here consider the spread of a disease within a population of N individuals, mod- elled by stochastic versions of the well-known SIR [3]. a susceptible state $S(t)$ and can then move to an infectious state $I(t)$ with an infection rate of $\beta$. These infectious individuals then move to a recovered state $R(t)$ with a recovery rate of $\gamma$, after which they can no longer be infected. The SIR model, governed by the state changes $S(t) \to I(t) \to R(t)$, thus has two model parameters $\boldsymbol{\theta}_1 = (\beta, \gamma)$.

The stochastic versions of these epidemiological processes are usually defined by a continuous-time Markov chain (CTMC), from which we can sample via the Gillespie algorithm [2]. However, this generally yields discrete population states that have undefined gradients. In order to test our gradient-based algorithm, we thus resort to an alternative simulation algorithm that uses stochastic differential equations (SDEs), where gradients can be approximated.

We first define population vectors $X_1(t) = (S(t), I(t))$ for the SIR model and $X_2(t) = (S(t), E(t), I(t))$ for the SEIR model. We can effectively ignore the population of recovered because the total population is fixed. The system of Itô SDEs for the above epidemiological processes is

$$\mathrm{d}\boldsymbol{X}(t) = \boldsymbol{f}(\boldsymbol{X}(t))\,\mathrm{d}t + \boldsymbol{G}(\boldsymbol{X}(t))\,\mathrm{d}\boldsymbol{W}(t), \tag{40}$$

where $\boldsymbol{f}$ is the drift term, $\boldsymbol{G}$ is the diffusion term and $\boldsymbol{W}$ is the Wiener process. Euler-Maruyama algorithm is used to simulate the sample paths of the above SDEs.

$$\boldsymbol{f}_{\text{SIR}} = \begin{pmatrix} -\beta \frac{S(t)I(t)}{N} \\ \beta \frac{S(t)I(t)}{N} - \gamma I(t) \end{pmatrix}, \boldsymbol{G}_{\text{SIR}} = \begin{pmatrix} -\sqrt{\beta \frac{S(t)I(t)}{N}} & 0 \\ \sqrt{\beta \frac{S(t)I(t)}{N}} & -\sqrt{\gamma I(t)} \end{pmatrix} \tag{41}$$

We use the infection rate ($I$) as 0.1 and the recovery ($R$) rate as 0.01. The independent priors are N(0.1,0.02) and N(0.01, 0.002). The initial infection number is 10. We update MI one time after updating sampler three steps.We use RNN network with 2 layer 64 dim hidden layer construction to decoder the sequential design.

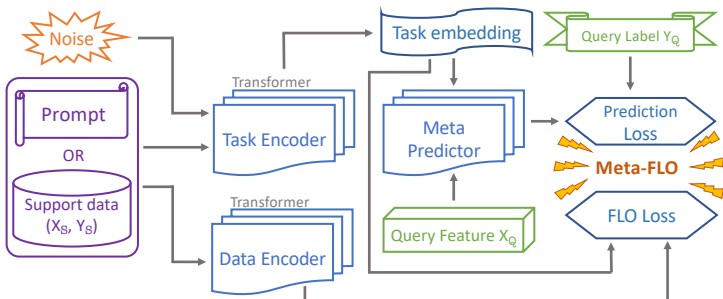

Figure S10: Model architecture of `Meta-FLO`.

## L    Meta Learning

**Intuitions.** Now let us describe the new `Meta-FLO` model for meta-learning. Given a model space $\mathcal{M}$ and a loss function $\ell : \mathcal{M} \times \mathcal{Z} \to \mathbb{R}$, the true risk and the empirical risk of $f \in \mathcal{M}$ are respectively defined as $R_t(f) \triangleq \mathbb{E}_{Z \sim \mu_t}[\ell(f, Z)]$ and $\hat{R}_t(f; \mathbb{S}_t) \triangleq \frac{1}{m} \sum_{i=1}^{m} \ell(f, Z_i)$. Let us denote $R_\tau$ is the generalization error for the task distribution $\tau$ where all tasks originate, and $\hat{R}_\tau$ is the empirical estimate. Our heuristic is simple, that is to optimize a tractable upper bound of the generalization risk given by

$$R_\tau \leq \underbrace{\hat{R}_\tau}_{\text{Utility}} + \underbrace{|R_\tau - \hat{R}_\tau|}_{\text{Generalization}} \triangleq \mathcal{L}_{\text{upper}}. \tag{42}$$

For meta-learning, we sample $n$-tasks for training and $n'$-tasks for testing, respectively denoted as $\mathbb{S}_{1:n}$ and $\mathbb{S}\text{test}_{1:n'}$. We further decouple the learning algorithm into two parts: the *meta-learner* $\mathcal{A}_{\text{meta}}(\mathbb{S}_{1:n})$ that consumes all train data to get the *meta-model* $f_{\text{meta}}$, and then *task-adaptation learner* $\mathcal{A}_{\text{adapt}}(f_{\text{meta}}, \mathbb{S}_t)$ which adapts the meta-model to the individual task data $\mathbb{S}_t$ to get task model $f_t$. For parameterized models such as deep nets, we denote $\Theta$ as our *meta parameters* and $E_t$ as *task-parameters*, that is to say $\Theta \triangleq \mathcal{A}_{\text{meta}}(\mathbb{S}_{1:n})$, $E_t \triangleq \mathcal{A}_{\text{adapt}}(\Theta, \mathbb{S}_t)$, where $\Theta, E_t$ can be understood as weights of deep nets. In subsequent discussions, we will also call $E_t$ the *task-embedding*. We can define the population *meta-risk* as $R_\tau(\Theta) \triangleq \mathbb{E}_{t, \Theta = \mathcal{A}_{\text{meta}}(\mathbb{S}_{1:n})}[\mathbb{E}_{E_t = \mathcal{A}_{\text{adapt}}(\Theta, \mathbb{S}_t)}[R_t(f_{E_t})]]$, and similarly for the empirical risk $\hat{R}_\tau$ evaluated on the query set $\mathbb{Q}_t$. Our model is based on the following inequality [4]:

$$\lim_{n \to \infty} |\mathbb{E}[R - \hat{R}]| \leq \sqrt{\frac{2\sigma^2}{m} I(E_t; \mathbb{S}_t | \Theta)} \tag{43}$$

which gives the main objective $\mathcal{L}_{\text{Meta-FLO}}(f) = \hat{R}(f) + \lambda \sqrt{I_{\text{FLO}}(\hat{\mathcal{D}}_t; \hat{E}_t)}$. We summarize our model architecture in Figure S10.

The sin-wave adaptation experiment involves regressing from the input ($x \sim \text{Uniform}([-5, 5])$) to the output of a sine wave $\kappa \sin(x - \gamma)$, where amplitude $\kappa \sim \text{Uniform}([0.1, 5])$ and phase ($\gamma \sim \text{Uniform}([0, \pi])$) of the sinusoid vary for each task. We use mean-squared error (MSE) as our loss and set the support-size = 3 and query-size = 2. We use simple three-layer MLPs for all the models: regressor, prompt encoder, and FLO critics, with hidden units all set to $[512, 512]$. During training, we use an episode-size of $64$. For MAML, we use the first-order implementation (FOMAML), and set inner learning rate to $\alpha = 10^{-4}$. For `Meta-FLO`, we set regularization strength to $\lambda = 10^{-2}$.