# OpenReview forum: "Tight Mutual Information Estimation With Contrastive Fenchel-Legendre Optimization"
_NeurIPS.cc/2022/Conference — NeurIPS 2022 Accept_

### Official Review · Reviewer_gbJf · 2022-06-24

**Rating:** 5
**Confidence:** 3
**Soundness:** 2 fair
**Presentation:** 2 fair
**Contribution:** 2 fair

**Summary:**

The paper proposes a new mutual information estimator based on Fenchel-Legendre optimization (FLO). The paper describes existing related techniques, and then presents the derivation of the FLO estimator via convex duality. The FLO estimate is established as a lower bound on the true mutual information value, and it is shown that appropriate choices of the g and u functions in the variational bound retrieves the true value. The authors also provide a convergence to stationarity claim for SGD-based optimization of the FLO objective. Numerical experiments comparing the proposed approach to existing estimators is provided, showing good performance on the considered examples. Applications to Bayesian optimal experiment design and meta-learning are also presented.

**Questions:**

* The feasibility claims from Proposition 2.3 and Corollary 2.4 are trivial and have limited utility since the optimal functions depend on the population distribution, which is unknown in practice.

* The abstract and intro make it sound like a statistical performance analysis for FLO will be provided, but it is nowhere to be found. The short paragraph after Corollary 2.4 is quite laconic and surely doesn't qualify as such. If there is a rigorous statement the authors can provide, it should be clearly formulated, stated, and proven. The current presentation is misleading in this regard.

* The convergence claim from Proposition 2.5 is local and should be framed as such from the get-go. The abstract and introduction make it sound like global guarantees would be provided, which is not the case and hence this presentation is misleading. The 2nd part of that statement that assumes convexity if I_{UBA} surely does not quality as a global guarantee since this convexity in the parameters is rarely available (e.g., when parametrizing the functions by a neural network, as often performed in practice).

* I find the statement in Table 1 that FLO has low bias and moderate variance quite odd. What actual evidence (beyond the said heuristics) did the authors provide to support this bold claim?

* In the `Efficient implementation of FLO' paragraph: jointly modeling u and g by a single neural network makes sense, but doesn't doing so typically require more parameters than approaches based on a single critic? If so, how can the authors say that this incurs no extra modeling cost?

* The bi-linear critic design proposed in the next paragraph is indeed highly parallelizable, but how expressive is this model? Under what conditions can we expect that class to be able to approximate the optimal u and g functions (that depend on the joint population distribution)? Computational efficiency is important, but it is crucial to understand the tradeoffs involved.

* I found the content of Section 2.3 full of jargon and quite confusing. I am not sure what the goal of this section is. The authors may consider omitting it to make space for a more comprehensive and rigorous treatment of the SGD convergence guarantees.

* It is unclear how FLO helps mitigate challenges associated with large sample sets and high-dimensional data. The text suggests that this is the case on several occasions, but I have seen no results to that effect. In a similar vein, the authors should contrast their approach with the McAllester and Stratos formal limitations paper [44].

* Equation (11) is unclear; in particular, StopGrad is never defined.

**Limitations:**

I don't think that limitations of the approach were sufficiently discussed. The paper advocates quite strongly that FLO is superior to existing approaches but provided evidence to that effect is at best partial. As mentioned above, these claims would have been easier to digest if supporting theory and formal guarantees was provided. Clarity on the limitations of the approach and/or when alternative methods are preferable would be appreciated as well.

**Strengths And Weaknesses:**

The proposed estimation approach is interesting and may very well have merits, but in my opinion, the paper does not provide sufficient concrete evidence to support that. I liked the overview of existing/past techniques, which was comprehensive and informative, and think that the FLO approach was clearly described. Beyond that, however, while there is a considerable amount of discussion and heuristic arguments for why FLO is favorable to alternative methods, these lack rigorous supporting evidence. In various cases in the past, an estimator of mutual information was proposed and argued to drastically improve upon past approaches only for the next paper to provide examples on which it fails. The best approach to avoid such pitfalls is to couple a proposed estimator with formal performance guarantees (e.g., consistency, error bounds, sample complexity, etc.), but the paper comes short in that regard. I also found the paper to be quite heavy in jargon and at times sloppy in its mathematical notation, which are aspects the authors should work to improve on. Specific concerns/comments/questions are provided in the next section.

---

> ### Author Response · Authors · 2022-08-02
> **Reply to Reviewer gbJf (part 2 of 2)**
>
> *Q: How expressive is the bi-linear approach?*
> * Good question. This is a topic of independent research interest and out of scope for the current study. We would like to point the reviewer to recent works such as [S4] for more discussions, where the author(s) of that paper find adding flexibility to the bi-linear critic improves performance.
>
> [S4] Zhang et al. Temperature as Uncertainty in Contrastive Learning. NeurIPS 2021, Self-Supervised Learning Workshop
>
> *Q: What’s the goal of Section 2.3? I found it full of jargon and quite confusing. Maybe it can be replaced with a more rigorous treatment of the SGD convergence.*
> * Thanks for your suggestion on the organization of this paper. To clarify, we hope Sec 2.3 can help expert readers better understand FLO’s scope and impact. It may appear a bit dense given the space limit and we will further iterate on its readability. Overall the current form is well received and we welcome any inputs that can make it even better. We have tried our best effort to make the technical contents rigorous and self-contained given the space limit.
>
> *Q: It is unclear how FLO helps mitigate challenges associated with large sample sets and high-dimensional data.*
> * Our key message is that FLO works better in high-MI, small batch-size settings compared to popular solutions. This is highly relevant for high-dimensional data, because model sizes are typically larger, and fewer training data points can be accommodated on the computing device in each batch. We will further emphasize this point based on this reviewer's feedback.
>
> *Q: The authors should contrast their approach with the McAllester and Stratos formal limitations paper [44].*
> * Our work fully acknowledges the fundamental limitations on unbiased nonparametric MI estimation outlined in McAllester et al., and we did discuss this paper in line 44. The fact the (slightly) biased MI estimator can do a much better job in variance control is well known in literature and is not unique to our solution. In fact, we showed our FLO pays less price in variance to achieve even better accuracy (Figure 3) when tight estimation is impossible. We will emphasize these points in our revision.
>
> *Q: StopGrad is never defined.*
> * This refers to the stop gradient operation used in popular deep learning frameworks. For example in PyTorch, this StopGrad is the detach() operation.
>
> *Q: The limitations of the approach were not sufficiently discussed*
> * While there is no dedicated section on the limitations in our manuscript, we did provide in-place discussions on potential issues such as extra modeling \& computation cost (e.g., line 171). We show for bigger models and with our suggested optimal implementation strategies these costs are either minimal or worthwhile. One thing we need to clarify is that we are not suggesting blindly swapping out estimators such as InfoNCE. There are scenarios where low-variance InfoNCE is still desired. As we stated in the paper, FLO is more favorable in the high-MI, small batch-size regime, where existing solutions are less competitive. We will make these points more clear in our revision.
>
> * Again, FLO complements the existing toolbox of contrastive mutual information estimation. In our own workflows, we usually go with InfoNCE/NWJ for quick prototyping and switch to FLO when working with more complex models or the bottleneck of batch size is felt. We also use different estimators for health check of model training as the g critic is the same for different estimators. Another interesting observation we have made in recent months is that with the more challenging Reinforcement Learning setups, FLO is more likely to quickly recover from training dips compared to InfoNCE. There are many empirical aspects like these not covered by theoretical analyses, and we are happy to share those with the community.
>
> *Q: The feasibility claims are trivial*
> * We note stating these results explicitly is still needed for the completeness of our work.
>
> *Q: The short paragraph after Corollary 2.4 doesn't qualify as statistical performance analysis.*
> * To clarify, that paragraph is to provide an intuitive argument that connects FLO to the InfoNCE. This is to help readers better understand our work, it is not a statistical performance analysis.

---

> > ### Comment · Reviewer_gbJf · 2022-08-05
> > **Answer after review**
> >
> > I thank the authors for their detailed answers and promised modifications. Let me mention that consistency and even sample complexity analyses can indeed be devised for variational estimators; cf. e.g.,
> > * https://arxiv.org/pdf/0809.0853.pdf for an analysis of a variational estimator of f-divergences; and
> > * https://www.jmlr.org/papers/volume23/21-1212/21-1212.pdf for a more recent analysis of neural estimators of f-divergences that also accounts for the approximation error (also covers MINE).
> >
> > I appreciate the impressive empirical performance of the proposed estimator and take into account the favorable responses of other reviewers. However, I believe that an accompanying theory, even if a mere consistency claim, would come a long way for a newly proposed method and significantly strengthen the paper (rather than just leaving it for future work). The tools for providing such an analysis should all be available.

---

> > > ### Author Response · Authors · 2022-08-05
> > > **More clarifications on the convergence theories mentioned by the reviewer**
> > >
> > > Thanks for following up our rebuttal. We are glad to see that now the reviewer better appreciates our work. We are inspired by the level of theoretical rigor the reviewer pursues, and find the mentioned references highly relevant. The work of [Nguyen, et al. 2009] is classical, and [Sreekumar, et al. JMLR 2022] is very recent and less known to the community. These two papers study the convergence theories for divergence estimators, which is more generic than MI. In our revision, we will discuss how to approach the convergence theories similar to [Nguyen, et al. 2010] and [Sreekumar, et al. JMLR 2022], and cite relevant works accordingly.
> > >
> > > However, we must point out that directly comparing our work to [Nguyen, et al. 2010] and [Sreekumar, et al. JMLR 2022] is not appropriate. We outline the major differences below.
> > >
> > > ### Empirical estimation versus oracle estimation
> > > Note that the convergence results in [Nguyen, et al. 2010] and [Sreekumar, et al. JMLR 2022] are derived based on oracle estimators such as $g_n^* = sup_{g \in \mathcal{G}} \int g d P_n - h(g) d Q_n$, that is to say, they are not concerned about how to actually implement this $\sup_{g \in G}$ operation for complex function spaces such as deep neural nets, where assumptions like $\mathcal{G}$ is convex ([Nguyen, et al. 2010]) are violated (to quote from [Srekkumar, et al. JMLR 2022], “This assumption is often violated in practice, e.g., when using a NN class as done herein, or a reproducing kernel Hilbert space (RKHS), as considered in [Nguyen, et al. 2010].” Also be mindful that the RKHS solution does not scale well to data size and complexity.).
> > >
> > > Our work cares more about the empirical estimation of mutual information with stochastic gradient descent (SGD) using neural nets, and we presented the convergence analysis under SGD. As such, we respectfully disagree with the evaluation that our work missed convergence theory. We want to emphasize that both the reviewer and us believe theoretical guarantees such as convergence are important. The convergence analyses presented by [Nguyen, et al. 2009] and [Sreekumar, et al. JMLR 2022] and the convergence analyses presented by us are complementary and they simply appeal to different audiences.
> > >
> > > ### Theoretical papers versus practical algorithms
> > > As an important remark, [Nguyen, et al. 2010] and [Sreekumar, et al. JMLR 2022] are pure theoretical investigations. Both took over 20 pages just to clarify the core theories. This is, however, not possible with the 9-page limit for a NeurIPS submission, and our dedication to clearly present a novel practical algorithm and show strong empirical evaluations to demonstrate the effectiveness for the community. In comparison, [Nguyen, et al. 2010] only presented simple 1-D toy model experiments, and [Sreekumar, et al. JMLR 2022] did not offer any empirical evaluations at all. In this paper we have strived to balance the theoretical rigor and accessibility within the page limit, so that more readers can benefit from our work, either practically or theoretically.
> > >
> > > **We believe machine learning is a beautiful marriage between maths and engineering, and we sincerely hope the readers can find such harmony in our paper.**
> > >
> > > Thank you.

---

> > > > ### Comment · Reviewer_gbJf · 2022-08-08
> > > > **Response**
> > > >
> > > > I thank the authors again for their response. I will reevaluate the paper, discuss with other reviewers, and consider adapting my score accordingly.

---

> > > > > ### Author Response · Authors · 2022-08-09
> > > > > **Thanks for the rebuttal feedback**
> > > > >
> > > > > Thank you for reconsidering our paper! Do not hesitate to let us know if you have any further questions or concerns, we are happy to clarify.

---

> ### Author Response · Authors · 2022-08-02
> **Reply to Reviewer gbJf (part 1 of 2)**
>
>
> Thanks for taking time reviewing our paper. See below for our replies for your comments.
>
> *Q: In various cases in the past, an estimator of mutual information was proposed and argued to drastically improve upon past approaches only for the next paper to provide examples on which it fails.*
> * We believe this is how science and technology evolve. For most problems, we can not feasibly hope to solve them in one-shot, finding the weakness of prior solutions is an integral part of scientific research.
>
> *Q: The best approach to avoid such pitfalls is to couple a proposed estimator with formal performance guarantees (e.g., consistency, error bounds, sample complexity, etc.), but the paper comes short in that regard.*
> * This is a good point, and we agree that proposed algorithms should ideally have formal performance guarantees. But our view is that this should not be a mandatory requirement otherwise it will stifle innovation. What usually happens is that the supporting theories for popular algorithms come years after. For example, Generative Adversarial Net (GAN) was founded on the heuristic that adversarial training optimizes Jensen-Shannon divergence. Its early formulations were notoriously fragile and did not come with any convergence guarantee, this does not prevent it from becoming one of the most popular machine learning techniques used today.
>
> * Also, we want to emphasize that not all frameworks admit analysis such as consistency, error bounds, and sample complexity. To the best of the author(s)' knowledge, variational inference techniques do not usually conform to such analytical frameworks. This is because variational techniques care more about tractable, efficient approximations, and the performance guarantees mentioned by the reviewer do not apply. Also, such analysis typically requires strong assumptions that are unverifiable in practice. To the best of our knowledge, the MIND work we mentioned does have some consistency analysis, but that work falls under a different theoretical framework so it is not an apple-to-apple comparison. Some classical MI estimators also have statistical guarantees, and we did provide fair empirical comparisons against those (including MIND).
>
> * Lastly, we note our work does provide some guarantees such as the (local) SGD convergence. No other MI estimation work has provided such analysis to the best of our knowledge.
>
> *Q: While there is a considerable amount of discussion and heuristic arguments for why FLO is favorable to alternative methods, these lack rigorous supporting evidence.*
> * We respectfully disagree with this assessment. The heuristic arguments help readers (non-experts in particular) better understand our work, and we do provide solid mathematical proofs justifying our FLO estimator. Our experiments also lend strong empirical support that FLO is a very competitive estimator, especially in the high-MI regime where other methods don’t work well (e.g., Figure 3).
>
> *Q: The convergence claim from Proposition 2.5 is local*
> * Yes, but we note that without strong assumptions like global convexity (as also mentioned by the reviewer, this usually does not hold in practice), all SGD convergence analyses are local. So this is not a limitation specific to our work and our narratives are consistent with prior literature on SGD convergence analyses. That said, we are happy to make this local convergence point more explicit in our revision based on the reviewer’s input.
>
> *Q: What actual evidence (beyond the said heuristics) did the authors provide to support the bold claim in Table 1 that FLO has low bias and moderate variance?*
> * This claim is based on empirical observations. For example in Figure 3, FLO is more accurate and less variable compared to other estimators. We also observe similar behaviors in other experiments. Prior variational MI estimation works have used numerical experiments as we show here to compare the variance, as there is no analytical framework that characterizes the variance theoretically.
>
> *Q: Jointly modeling u and g by a single neural network makes sense, but doing so typically requires more parameters. So how can the authors say that this incurs no extra modeling cost?*
> * Take the popular ResNet-50 encoder as an example. The ResNet has about 24M parameters, with the last layer outputs feature of dimension 2,048. So the difference between a single $g$ and a joint $(g, u)$ networks are 24M+2k and 24M+2*2k, which is 0.008\% difference in terms of parameters. That said, we favor the bi-linear parameterization described in line 171 over this joint parameterization, as the bi-linear one is more parallelizable.

---

> ### Comment · Reviewer_NZCL · 2022-08-09
> **Comment from Reviewer NZCL**
>
> Reviewer NZCL here...
>
> I have reviewed the discussion in this thread and it seems that the strongest criticism of reviewer gbJf is the lack of a detailed theoretical analysis.  In my view, this paper is largely practical and provides a reasonable basis of theoretical justification.  I agree with the author response that the level of rigor required to analyze a highly nonconvex optimization is prohibitive and not likely to yield strong consistency results.  I also agree that this should not prohibit the development of innovative methods, the authors turn to GANs as an excellent example of this.
>
> I would encourage reviewer gbJf to consider whether the insistence on strong theoretical analysis is a reasonable baseline for publication.

---

### Official Review · Reviewer_4a7X · 2022-07-10

**Rating:** 6
**Confidence:** 3
**Soundness:** 3 good
**Presentation:** 3 good
**Contribution:** 3 good

**Summary:**

This paper introduces new type of contrastive mutual information estimator, which is tight and provides theoretical convergence. As the variance of previous estimator was big trouble for the accurate MI estimation, it provides new optimization technique with fenchel-legendre optimization. The relationship with the existing mutual information estimator has been nicely provided, which makes it easier to evaluate the introduced estimator with the fair manner.

**Questions:**

Q1. how this framework can be applied on the supervised contrastive learning? Additional works from supervised contrastive learning stated that it is connected with conditional mutual information maximization. I am curious that the mutual information estimation or maximization given the label can be conducted with FLO.


**Limitations:**

Stated in weakness session.

**Strengths And Weaknesses:**

Strengths:

Theoretical analyses : it provides the tightness and boundedness for the given estimator with nice derivation. It was easy to follow.
Experiments : FLO significantly outperformed its variational counterparts in the more challenging 263 high-MI regime, which implies the robustness from the noisy environments.

Weaknesses : Experiments for additional data-efficient learning is not provided. The author referred that efficient implementation for FLO can be provided, which makes it easier to apply FLO on data-efficient learning. However, it was not adequately managed in the manuscripts.

---

> ### Author Response · Authors · 2022-08-02
> **Reply to Reviewer 4a7X**
>
> Thanks for the positive feedback! Your comments are carefully addressed below.
>
> *Q: Comments regarding data-efficient learning.*
> * Thanks for the input. Our understanding is that the reviewer is asking about two separate issues: the efficient implementation of FLO, and the data-efficient learning experiments.
>
> * For the efficient implementation of FLO, we have described the fast bi-linear algorithm in line 171-187. In practice, users can simply call a function ```FLO_Loss(x_feature, y_feature)``` to use FLO without worrying about implementation details. Our supplementary material contains code examples.
>
> * Our work explored multiple directions where FLO can help with data-efficient learning. A straightforward application is self-supervised pretraining, which is framed under the multi-view representation learning experiment in our manuscript. The other two directions are relatively new to the NeurIPS community because they are emerging subjects: One is to collect more informative data using Bayesian Optimal Experiment Design so that the downstream application can accurately make an inference with fewer data. The second is a novel formulation of information-theoretic meta/few-shot learning so that the backbone model can adapt to new tasks with fewer supervision (i.e., using less annotated data). The setups for these two approaches are less well known, and we will add extra clarifications and more experiment results in our revision as suggested by the reviewer, space allowed. Thanks for making this suggestion.
>
> *Q: Can this be applied to supervised contrastive learning?*
> * Yes, absolutely. Most self-supervised contrastive learning optimizes ```MI(X; X’)```, where $X, X’$ are two random augmentations of the same data point, while supervised contrastive learning optimizes ```MI(Z; Z’)```, where $Z, Z’$ are augmentations of two data points with the same label (some literature interpret it as the conditional MI). You can simply swap out the InfoNCE loss used in supervised contrastive learning with our FLO loss.

---

### Official Review · Reviewer_FJTi · 2022-07-10

**Rating:** 8
**Confidence:** 2
**Soundness:** 4 excellent
**Presentation:** 4 excellent
**Contribution:** 3 good

**Summary:**

The paper introduces FLO, a novel contrastive estimator for mutual information. FLO is inspired by a reexamined connection between mutual information and unnormalised statistical modelling, which yields a unified theoretical basis for several variational-based MI estimators. FLO is then proven to be tight, and to converge under SGD. Finally, several experiments confirm FLO's superior performance compared to previous estimators.

**Questions:**

My main suggestions for improvements would go for the experimental section:
- running time comparisons
- giving more context as to levels of performance for various tasks.

**Limitations:**

Limitations are addressed adequately

**Strengths And Weaknesses:**

The paper is *very* well written, and as a consequence a pleasure to read, even for non-experts in the field of mutual estimation. Connections are explained clearly and the new insights and methods are well-motivated. The 'zoo' of estimators in Figure 1 is much appreciated.

I can find precious little weaknesses in the paper (although I'm by no means an expert in this particular field). A few things:
- some running time experiments contrasting the performance of the various estimators would be good, to verify the tractability claims
- in the experiments, some explanation of how significant the gains are would be good, given that some of the tasks are not familiar to all readers
- all told, given the amount of material in the paper (and consequently how much of it is left in the appendix), I wonder if a journal version would not do it better justice.

---

> ### Author Response · Authors · 2022-08-02
> **Reply to Reviewer FJTi**
>
> Thanks for the positive review and constructive inputs! We have clarified your questions below.
>
> *Q: Some running time experiments to verify the claims*
> * FLO only incurs minor computation overhead compared to existing estimators, and for a large models, the increase in computation time is almost neglectable. Below we have attached the runtime comparison for the standard BERT (```BertModel.from_pretrained("bert-base-uncased")```) and ResNet50 (```torch.hub.load("pytorch/vision", "resnet50")```) on a V100 32G GPU for each training iteration, and the difference is neglectable.
>
> **Table S1. Runtime comparison between FLO and InfoNCE (unit s)**
>
> | ReseNet50 | bs = 16 | bs = 32 | bs = 64 | bs = 128 |
> --- | --- | ---| ---| ---|
> | FLO | 0.107 | 0.202 | 0.441 | 0.74 |
> | InfoNCE | 0.106 | 0.201 | 0.423 | 0.731 |
>
> | BERT | bs = 16 | bs = 32 |
> --- | --- | ---|
> | FLO | 0.724 | 1.356 |
> | InfoNCE | 0.718 | 1.343 |
>
> *Q: Explanation of how significant the gains given some of the tasks are not familiar to all readers*
> * Thanks for this suggestion, we will give more context to the tasks in our revision. The bias-variance tradeoff plots are the standard benchmark for the MI estimators. Based on our results, FLO fares well in the high MI regime whereas other estimators struggle. This feature will unlock applications requiring accurate high-MI estimates. For example, the meta-learning experiment is a novel application of mutual information estimation in data-efficient learning enabled by FLO. To the best of our knowledge, this is the first practical algorithm that directly optimizes the information-theoretic generalization bound; this alone is a significant contribution. The Bayesian Optimal Experiment Design results will appeal to statisticians: we show FLO consistently outperforms alternatives in our experiments, while in prior literature there is no consensus on which estimator works best (our experiments reproduced such observation).
>
> *Q: A journal version might do better justice for the paper*
> * Thanks for this suggestion, we do have plans for an extended journal version. Given the rapid pace of the field, top-tier venues such as NeurIPS help us to share our findings with the community in a more timely manner. This work is an integral part of a more comprehensive data-efficient learning research thread we are currently building (beyond MI-based solutions), at some point in the future we will consolidate our findings into a long journal version.

---

> > ### Comment · Reviewer_FJTi · 2022-08-08
> > **thanks**
> >
> > Thank you for your response which clears up most of the questions I had about the paper.
> > The time comparisons are convincing, so I will update my score accordingly.

---

> > > ### Author Response · Authors · 2022-08-09
> > > **Thanks for the rebuttal feedback**
> > >
> > > Thank you for the positive response! We are glad that our rebuttal helped to clarify your questions. If you have any further questions or concerns, please do not hesitate to let us know.
> > >
> > > Thanks

---

### Official Review · Reviewer_NZCL · 2022-07-12

**Rating:** 8
**Confidence:** 5
**Soundness:** 4 excellent
**Presentation:** 4 excellent
**Contribution:** 3 good

**Summary:**

The paper proposes a new bound of mutual information (MI) based on Fenchel-Legendre Optimization (FLO).  The bound has a number of beneficial properties such as tightness (it can approach the true MI), the empirical estimator is unbiased, and it can be estimated using samples from the model but does not require evaluation of model probabilities (i.e. it can be used for implicit likelihood models).  The paper discusses properties of the FLO estimator in relation to existing variational estimators and those based on the conjugate function.  Finally , the paper provides theoretical convergence guarantees along with empirical analysis (and ablation study) on several interesting models that are relevant in the literature on MI estimation and sequential experiment design.

**Questions:**

* Proposition 2.2 states that FLO is a lower bound on UBA with the same critic.  If so, what is the benefit of FLO over UBA (the unbiased empirical estimator is obviously one, but are there others)?  Is this bound true for every $u_\phi$ or only for an optimal one?

* Several estimators (e.g. DV) are based on a similar conjugate duality construction to FLO.  Yet, the authors do not discuss the relationship between these estimators in great detail.  How do these estimators compare, e.g. for similar choices of the surrogate function (i.e. the critic in the FLO case)?



**Limitations:**

The authors clearly state limitations.  This reviewer does not see obvious negative societal impact of the work.

**Strengths And Weaknesses:**

**STRENGTHS**
The paper is very well-written, the problem is well-motived, and the topic is timely and important.  The authors provide a solid foundation for the reader, along with all necessary context.  The paper is a good blend of theory and practical application.  The empirical analysis provides comparison to most relevant MI estimators and sufficiently supports the stated claims.

One strength that receives less attention due to the focus on tightness (see critique below) is that the empirical estimator lacks finite sample bias.  Much of the literature on MI estimation focuses on properties of the analytic approximations / bounds, when in reality these bounds cannot be calculated exactly and require further estimation.  In most cases there do not exist unbiased estimators, for example in the UBA bound, and estimators are only asymptotically unbiased / consistent.  By contrast, FLO admits a straightforward empirical estimator with zero finite sample bias.  Discussion of this is provided in Zheng et al. (2018) and Rainforth et al. (2018) for the Barber and Agakov bound.

**WEAKNESSES**
The empirical analysis in the main text is fairly superficial.  Results on the SIR model are particularly cursory, and are largely qualitative.  For example, the caption of Figure 6 states that FLO designs are somehow more useful because they are sampled more densely around the infection spike.  Yet the authors provide no comparison and the comment is debatable from visual inspection alone.  Additionally, the estimates provided in Figure 3 are mostly qualitative and do not seem to show a strong benefit of FLO compared to NWJ for example.

The paper leans heavily on tightness of the FLO estimator, yet this property does not hold in practice.  The discussion under Prop. 2.3 shows that tightness is achieved only under the optimal critic $g_{*}(x,y)$ which requires knowledge of the true posterior $p(x\mid y)$.  In addition, the tightness property is not unique to FLO, despite what the authors suggest in Table 1.  Many of the provided estimators are also asymptotically tight or tight given knowledge of the true posterior.  For example, most variational MI bounds become tight when the variational distribution $q(x \mid y) = p(x \mid y)$ almost everywhere.  By Prop. 2.2 the UBA bound is also tight given the same optimal critic, and indeed FLO is in general a looser bound than UBA by this proposition.   Similar properties hold for DV and NWJ estimators as well as InfoNCE (i.e. Proposition 2.1).

Some detailed comments below:
* It is not clear what point the authors intend to convey with the section on unnormalized statistical modeling and its connection to MI estimation.  Essentially any posterior inference problem contends with the lack of an analytic partition function, and so it is not particular to this setting.
* The statement in Prop. 2.3 is a bit terse.  At the very least authors should state that the estimator is tight at $(g_{*}, u_{g_{*}})$.
* Line 266 : Change “direct out attention” to “direct our attention”

**REFERENCES**

Rainforth, Tom, et al. "On nesting monte carlo estimators." International Conference on Machine Learning. PMLR, 2018.

Zheng, Sue, Jason Pacheco, and John Fisher. "A robust approach to sequential information theoretic planning." International Conference on Machine Learning. PMLR, 2018.

---

> ### Author Response · Authors · 2022-08-02
> **Reply to Reviewer NZCL**
>
> Thanks for the very positive comments! Please find our point-by-point response below.
>
> *Q: Benefit of FLO compared to NWJ?*
> * Figure 3 shows that FLO is more accurate and stable in the high MI regime, and the difference is more pronounced in the high-MI portion (i.e., $>8$ nats, see the figure below for detailed comparison). Also, in Figure 4, we can see that FLO consistently outperforms NWJ. Figure 4(b) also showed that in practice NWJ is not robust enough and sometimes needs additional care like a warm-start to get it working, that is the reason NWJ is rarely used in large-scale learning. Similar observations of NWJ have been reported in prior literature (e.g., SMILE [S1], CLUB [S2]).
>
> (If the figure is not showing up correctly, you can directly use the anonymous image link)
> ![Zoomed in comparison of FLO and NWJ in the high MI regime](https://i.ibb.co/994NtrZ/flo-nwj.jpg)
>
> [S1] Song, J. et al. Understanding the limitations of variational mutual information estimators. ICLR, 2020
>
> [S2] P. Cheng, et al., “CLUB: A contrastive log-ratio upper bound of mutual information,” ICML, 2020.
>
> *Q: Comment on the tightness property (not unique to FLO, and in practice true posterior is not available)*
> * We have emphasized the tightness property in this work because:
>   1. To contrast recent literature (e.g., InfoNCE [S3], SMILE [S1], etc.) which has advocated the use of biased MI estimators, where the gains from stability outweigh the loss in accuracy;
>   2. FLO’s empirical estimator admits zero finite sample bias.
>
> * For high MI applications, the FLO estimator may offer better bias-variance trade-offs compared to existing works. Our work shows when the exact MI is unrecoverable (i.e., true posterior hard to approximate), our FLO estimator promises to give a more accurate estimate with a smaller variance compared to state-of-the-art alternatives.
>
> [S3] Poole, B., et al. On variational bounds of mutual information. ICML 2019.
>
> *Q: Why mention unnormalized statistical modeling? Lack of partition function is common in posterior inference*
> * We approached our FLO solution from the perspective of energy modeling, not from posterior modeling -- and in the paper we presented our original thought process. Thanks for the constructive comment, we will add discussions on posterior inference in our revision.
>
> *Q: What’s the benefit of FLO over UBA? Is the bound true for all $u_{\phi}$?*
> * In practice, UBA is implemented with empirical estimators such as DV or MINE. However, DV is notoriously unstable, and its smoothed version MINE requires additional tuning to work well. In contrast, FLO works out-of-box as InfoNCE and NWJ. This bound holds for all $u_{\phi}$.
>
> *Q: Relations between estimators in the conjugate family*
> * Short answer is that the duality trick is applied to different (equivalent) formulations of MI. We will add a section in the Appendix to give more details.
>
> *Misc*
> * Thanks for picking up the typo, making suggestions on Prop 2.2, and bringing up related work Zheng et al. (2018) and Rainforth et al. (2018) ! We have updated the manuscript accordingly based on these inputs.

---

### Author Response · Authors · 2022-08-02
**Thanks for reviewing our work**

We appreciate all reviewers who have taken their time reviewing our work. We are encouraged by the very positive feedback and want to thank for the constructive inputs that help us further improve this paper. We have carefully prepared the rebuttal and please find our point-to-point responses to each individual comment after each review. Feel free to ask more follow-up questions or request more details, we are more than happy to address any lingering questions of our work in the interactive author-reviewer discussion.

---

### Public Comment · ~Artem_Sobolev1 · 2023-05-10
**FLO is a lower bound on NWJ**

__UPD__: see the corrected derivation in my next comment

---

Recall that the classic NWJ lower bound of MI has the form
$$
I\_\text{NWJ}
= \mathop{\mathbb{E}}\_{p(x,y)} f(x,y) - \mathop{\mathbb{E}}\_{p(x,y)} f(x,y) + 1
$$

The FLO lower bound as presented in eq. 8 has the form
$$
I\_\text{FLO} =
\mathop{\mathbb{E}}\_{p(x,y)} \left[ -u(x, y) - e^{-u(x, y)} \mathop{\mathbb{E}}\_{p(y')} e^{g(x, y') - g(x,y)} \right] + 1
$$

Or, slightly reformulated
$$
I\_\text{FLO} =
\mathop{\mathbb{E}}\_{p(x,y)} \left[ -u(x, y) \right] - \mathop{\mathbb{E}}\_{p(x, y)} e^{-g(x,y)-u(x, y)} \mathop{\mathbb{E}}\_{p(x')p( y')} e^{g(x', y')} + 1
$$

First, one can already see striking similarities. Next, see the Proposition 2.3 from the paper: the optimal $g^\star(x,y) = -u^\star(x,y) + \log p(x) + c(x)$ for arbitrary $c(x)$ and $u^\star(x,y) = \log p(x) - \log p(x|y)$. Let's substitute $g^\star$ and treat $I_\text{FLO}$ as a function of $u^\star$ and $c$:
$$
I\_\text{FLO}[u^\star, c] =
\mathop{\mathbb{E}}\_{p(x,y)} \left[ -u^\star(x, y) \right] - \mathop{\mathbb{E}}\_{p(x)} e^{\log p(x) + c(x)} \mathop{\mathbb{E}}\_{p(x')p( y')} e^{-u^\star(x,y) - c(x) - \log p(x)} + 1
$$

Next, let's reparametrize $f(x,y) = -u^\star(x, y)$ and set $c(x) = -\log p(x) + \gamma$ [1] to get
$$
I\_\text{FLO}[f, c] =
\mathop{\mathbb{E}}\_{p(x,y)} \left[ f(x, y) \right] - \mathop{\mathbb{E}}\_{p(x')p( y')} e^{f(x,y)} + 1
$$

Which is the NWJ bound. That is, we took $I_\text{FLO}$ lower bound with two arbitrary critics $g$ and $u$, optimized one out (if you fix $u$ and solve for optimal $g$ you'd get $g^\star(x,y) = -u(x,y) + \log p(x) + c$) and obtained the $I_\text{NWJ}$ bound. Hence the former is a lower bound on the latter.



[1] The narrowing assumption I made is that $c(x)$ is $-\log p(x)$ up to a constant $\gamma$ on the support of $p(x)$. The paper claims that _any_ $c(x)$ should work, but does not substantiate this claim. However, in order to cancel $c(x)$ out for the optimal critic(s) we need $\mathbb{E}\_{p(x)} e^{-c(x)-\log p(x)} \mathbb{E}\_{p(x')} e^{c(x')+\log p(x')}$ to be equal to 1, which surely doesn't hold for any $c(x)$.

---

> ### Public Comment · Authors · 2023-06-18
> **Response**
>
> Thank you for your comments. We have carefully gone through your post, but we disagree with your conclusion. Please see details below.
> In your derivations, one of the key step is that you went from the second equation in the post
> $$I_{FLO} =\underset{p(x,y)}{\mathbb{E}}[-u(x,y)-e^{-u(x,y)}\underset{p(y^{'})}{\mathbb{E}}e^{g(x,y^{'})-g(x,y)}]+1$$
> to the third equation in the post
> $$I_{FLO} = \underset{p(x,y)}{\mathbb{E}}[-u(x,y)]-\underset{p(x,y)}{\mathbb{E}}e^{-g(x,y)-u(x,y)}\underset{{p(x^{'})p(y^{'})}}{\mathbb{E}}e^{g(x^{'},y^{'})}+1$$
> However, this step is incorrect as in general,
> $$\underset{p(x,y)}{\mathbb{E}}[e^{-u(x,y)}\underset{p(y^{'})}{\mathbb{E}}e^{g(x,y^{'})-g(x,y)}]=\underset{p(x,y)}{\mathbb{E}}[e^{-u(x,y)-g(x,y)}\underset{p(y^{'})}{\mathbb{E}}e^{g(x,y^{'})}]\neq \underset{p(x,y)}{\mathbb{E}}e^{-g(x,y)-u(x,y)}\underset{p(x^{'})p(y^{'})}{\mathbb{E}}e^{g(x^{'},y^{'})}$$
> To see this more clearly, let us denote $h(x,y)=e^{-u(x,y)-g(x,y)}$ and $f(x)=\underset{p(y^{'})}{\mathbb{E}}e^{g(x,y^{'})}$ to simplify the notations, then it is easy to see
> $$\underset{p(x,y)}{\mathbb{E}}[h(x,y)f(x)]\neq\underset{p(x,y)}{\mathbb{E}}[h(x,y)]\underset{p(x)}{\mathbb{E}}[f(x)]$$

---

> > ### Public Comment · ~Artem_Sobolev1 · 2023-08-12
> > **Yes, but the claim still holds**
> >
> > Yes, thank you for noticing the typo. It is in fact important that two $x$ are the same.
> >
> > ---
> >
> > However, the claim still holds, here's the corrected derivaion:
> >
> > $$
> > I\_\text{NWJ}
> > = \mathop{\mathbb{E}}\_{p(x,y)} f(x,y) - \mathop{\mathbb{E}}\_{p(x,y)} \exp(f(x,y)) + 1
> > $$
> >
> > The FLO lower bound as presented in eq. 8 has the form
> > $$
> > I\_\text{FLO} =
> > \mathop{\mathbb{E}}\_{p(x,y)} \left[ -u(x, y) - \exp(-u(x, y)) \mathop{\mathbb{E}}\_{p(y')} \exp(g(x, y') - g(x,y)) \right] + 1
> > $$
> >
> > Or, slightly reformulated
> > $$
> > I\_\text{FLO} =
> > \mathop{\mathbb{E}}\_{p(x,y)} \left[ -u(x, y) \right] - \mathop{\mathbb{E}}\_{p(x, y)} \left[ \exp(-g(x,y)-u(x, y)) \mathop{\mathbb{E}}\_{p( y')} \exp(g(x, y')) \right] + 1
> > $$
> >
> > Now I'm going to show that one can reparametrize the critics $g$ and $u$ to arrive at the NWJ bound. In particular, see the Proposition 2.3 from the paper: the optimal $ g^\star(x,y) = -u^\star(x,y) + \log p(x) + c(x) $ for arbitrary $c(x)$ and $u^\star(x,y) = \log p(x) - \log p(x|y)$. Let's fix $u(x,y)$, assume we've optimized $g$ to the optimum, substitute $g^\star$ and treat $I_\text{FLO}$ as a function of $u^\star$ and $c$:
> > $$
> > I\_\text{FLO}[u^\star, c] =
> > \mathop{\mathbb{E}}\_{p(x,y)} \left[ -u^\star(x, y) \right] - \mathop{\mathbb{E}}\_{p(x, y)} \left[\exp(-\log p(x) - c(x)) \mathop{\mathbb{E}}\_{p(y')} \exp(-u^\star(x,y') + c(x) + \log p(x)) \right] + 1
> > $$
> > $$
> > I\_\text{FLO}[u^\star, c] =
> > \mathop{\mathbb{E}}\_{p(x,y)} \left[ -u^\star(x, y) \right] - \mathop{\mathbb{E}}\_{p(x) p(y)} \exp(-u^\star(x,y)) + 1
> > $$
> >
> > Which is the NWJ bound with $f(x,y) = -u^\star(x, y)$. That is, we took $I_\text{FLO}$ lower bound with two arbitrary critics $g$ and $u$, optimized one out and obtained the $I_\text{NWJ}$ bound. Hence the former is a lower bound on the latter.

---

### Meta-Review · Area_Chair_YriA · 2022-08-24

**Recommendation:** Accept
**Confidence:** Certain

**Metareview:**

All reviewers agree that the paper proposes an interesting and novel bound of mutual information (MI) based on Fenchel-Legendre Optimization. Although some reviewers have some technical concerns at their first reviews, basically those have been resolved by the authors' responses. Thus, although there are some points that should be modified from the current form, I think we can expect the authors modify the paper in the camera-ready by reflecting the discussion. Based on these, I recommend acceptance for this paper.

**Award:**

No

---

### Decision · Program_Chairs · 2022-09-14

Accept